# Fair Kernel K-Means: from Single Kernel to Multiple Kernel

**Peng Zhou** *
School of Computer Science and Technology
Anhui University
Hefei, 230601
zhoupeng@ahu.edu.cn

**Rongwen Li**
School of Computer Science and Technology
Anhui University
Hefei, 230601
e22301284@stu.ahu.edu.cn

**Liang Du**
School of Computer and Information Technology
Shanxi University
Taiyuan, 237016
duliang@sxu.edu.cn

## Abstract

Kernel k-means has been widely studied in machine learning. However, existing kernel k-means methods often ignore the *fairness* issue, which may cause discrimination. To address this issue, in this paper, we propose a novel Fair Kernel K-Means (FKKM) framework. In this framework, we first propose a new fairness regularization term that can lead to a fair partition of data. The carefully designed fairness regularization term has a similar form to the kernel k-means which can be seamlessly integrated into the kernel k-means framework. Then, we extend this method to the multiple kernel setting, leading to a Fair Multiple Kernel K-Means (FMKKM) method. We also provide some theoretical analysis of the generalization error bound, and based on this bound we give a strategy to set the hyper-parameter, which makes the proposed methods easy to use. At last, we conduct extensive experiments on both the single kernel and multiple kernel settings to compare the proposed methods with state-of-the-art methods to demonstrate their effectiveness. Our code is available at `https://github.com/rongwenli/NeurIPS24-FMKKM`.

## 1 Introduction

Clustering is a fundamental unsupervised machine learning task. In clustering, kernel methods, such as Kernel K-Means (KKM), can effectively separate nonlinear data into different clusters. Therefore, KKM has been widely studied in both the single kernel setting and multiple kernel setting [39, 50, 14, 15].

Notice that, in real-world applications, clustering is often used in some scenarios involving humans such as social networks [36] and crime analysis [32]. In these scenarios, since the humans are involved, we should guarantee the *fairness* of the clustering result, so that the clustering result will not cause discrimination to some specific groups. In the clustering task, we often consider the *group fairness*, where we have some pre-given groups that may suffer from the potential discrimination, called *protected groups*. Group fairness aims to partition data into some clusters and guarantee that no clusters contain a disproportionately small or large number of data in some specific protected

---

*Peng Zhou is the corresponding author. Peng Zhou and Rongwen Li are also with Anhui Provincial International Joint Research Center for Advanced Technology in Medical Imaging.

groups [6]. Although the above-mentioned kernel k-means and multiple kernel k-means methods show promising performance in the clustering task, none of them considers the fairness issue, and thus they may obtain some clustering results which cause discrimination to some groups.

To tackle this problem, in this paper, we propose a novel fair kernel k-means method and extend it from the single kernel setting to the multiple kernel setting. We follow a widely-used definition of fairness defined in [6], which is shown as Definition 1. By analyzing this definition, we carefully design a new fairness regularization term and prove that minimizing this term can lead to the optimal fairness defined in [6]. Besides, we observe that our fairness regularization term has a similar form of the loss function of KKM, and thus can be naturally and seamlessly plugged into the KKM framework, yielding an extremely simple and elegant Fair Kernel K-Means (FKKM) framework. This framework is so concise that we do not even need to modify the loss of KKM but just adjust the input kernel to our proposed *fair kernel*. This framework can also be easily extended to the Multiple Kernel K-Means (MKKM) task, leading to Fair Multiple Kernel K-Means (FMKKM). We also provide some theoretical analysis of its generalization error bound. Furthermore, based on the generalization error bound, we provide a strategy to set the hyper-parameter in our framework, which makes the method easy to use. Extensive experiments on single kernel clustering and multiple kernel clustering tasks show the effectiveness of our framework w.r.t. both the clustering accuracy and fairness.

The main contributions of our paper are summarized as follows:

- We propose a novel fairness regularization term and prove that minimizing this term can reach the optimal fairness defined in [6].
- Our proposed regularization term has a similar form to the KKM, and thus can be seamlessly integrated into the KKM and MKKM framework. To the best of our knowledge, this is the first work for fair kernel k-means and fair multiple kernel k-means.
- We provide a strategy to set the hyper-parameter based on the theoretical analysis, which makes the methods easy to use.
- Extensive experiments in both single and multiple kernel clustering show the effectiveness and superiority of our proposed methods compared with the state-of-the-art methods.

## 2 Related Work and Preliminaries

In this paper, we use a bold uppercase letter (e.g. $\mathbf{M}$) and a bold lowercase letter (e.g. $\mathbf{v}$) to denote a matrix and a vector, respectively. Given a matrix $\mathbf{M}$, we use $M_{ij}$ to denote its $(i, j)$-th element.

### 2.1 Kernel K-means and Multiple Kernel K-means

Given a data matrix $\mathbf{X} = [\mathbf{x}_1, \dots, \mathbf{x}_n] \in \mathbb{R}^{d \times n}$ with $n$ instances and $d$ features, let $\Phi(\cdot) : \mathbb{R}^d \mapsto \mathcal{H}$ represents a kernel mapping that maps $\mathbf{X}$ into a Reproducing Kernel Hilbert Space (RKHS) $\mathcal{H}$. The objective function of the kernel k-means with the sum-of-squares loss can be written as [39, 24]:

$$\min_{\mathbf{M}, \mathbf{Y} \in Ind} \|\Phi(\mathbf{X}) - \mathbf{M}\mathbf{Y}^T\|_F^2, \tag{1}$$

where $\Phi(\mathbf{X}) = [\Phi(\mathbf{x}_1), \dots, \Phi(\mathbf{x}_n)]$ and $\mathbf{M} = [\mathbf{m}_1, \dots, \mathbf{m}_c]$ represents $c$ clustering centroids in the RKHS $\mathcal{H}$. $\mathbf{Y} \in \{0, 1\}^{n \times c}$ is an indicator matrix, which is denoted as $Ind$, and $Y_{ij} = 1$ if $\mathbf{x}_i$ is assigned to the $j$-th cluster, and otherwise $Y_{ij} = 0$. Setting the derivative of Eq.(1) w.r.t. $\mathbf{M}$ to zero, we can obtain the closed-form solution of $\mathbf{M}$. Taking it back to Eq.(1), it can be rewritten as [42]:

$$\min_{\mathbf{Y} \in Ind} \text{Tr}(\mathbf{K}) - \text{Tr}\left(\left(\mathbf{Y}^T\mathbf{Y}\right)^{-\frac{1}{2}} \mathbf{Y}^T \mathbf{K} \mathbf{Y} \left(\mathbf{Y}^T\mathbf{Y}\right)^{-\frac{1}{2}}\right), \tag{2}$$

where $\mathbf{K} = \Phi(\mathbf{X})^T \Phi(\mathbf{X}) \in \mathbb{R}^{n \times n}$ is a kernel matrix with $K_{ij} = \Phi(\mathbf{x}_i)^T \Phi(\mathbf{x}_j)$. For the convenience of optimization, we denote $\mathbf{H} = \mathbf{Y}\left(\mathbf{Y}^T\mathbf{Y}\right)^{-\frac{1}{2}}$. Since directly solving Eq.(2) is an NP-hard problem [16], previous works [26, 41, 21] substituted the constraints $\mathbf{Y} \in Ind$ with $\mathbf{H}^T\mathbf{H} = \mathbf{I}$, leading to:

$$\min_{\mathbf{H}^T\mathbf{H}=\mathbf{I}} \text{Tr}\left(\mathbf{K}\left(\mathbf{I} - \mathbf{H}^T\mathbf{H}\right)\right). \tag{3}$$

The optimal $\mathbf{H}$ is formed by the $c$ eigenvectors of $\mathbf{K}$ corresponding to the $c$ largest eigenvalues. After obtaining $\mathbf{H}$, existing methods [54, 44, 35, 17] learn the final clustering results through some post-processing techniques such as k-means or spectral rotation on $\mathbf{H}$.

Multiple kernel k-means aims to fuse multiple base kernels to a consensus one for kernel k-means. Previous works assume that the ideal consensus kernel matrix is a combination of base kernel matrices i.e., $\mathbf{K}^* = \sum_{p=1}^m \gamma_p^2 \mathbf{K}^{(p)}$, where $\mathbf{K}^*$ is the consensus kernel matrix, and $\mathbf{K}^{(p)}$s are base kernels [27, 28, 19]. $\gamma_p$ is the weight of the $p$-th base kernel. Replacing $\mathbf{K}$ in Eq.(3) with the consensus kernel $\mathbf{K}^*$, we can obtain the objective function of MKKM:

$$\min_{\mathbf{H},\boldsymbol{\gamma}} \mathrm{Tr}\left(\mathbf{K}^*\left(\mathbf{I} - \mathbf{H}^T\mathbf{H}\right)\right), \ s.t. \ \mathbf{H}^T\mathbf{H} = \mathbf{I}, \ \boldsymbol{\gamma}^T\mathbf{1} = 1, \ \gamma_p \geq 0, \ \mathbf{K}^* = \sum_{p=1}^m \gamma_p^2 \mathbf{K}^{(p)}. \tag{4}$$

It can be solved by alternatively optimizing $\mathbf{H}$ and $\boldsymbol{\gamma}$.

## 2.2 Fair Clustering

Fair clustering considers the fairness in the clustering, which is an important problem in unsupervised machine learning. It was first introduced by Chierichetti et al., who proposed a fair decomposition method to avoid all members of a protected group being clustered into the same cluster [9]. However, this method can only handle two protected groups. To tackle this problem, Bera et al. further proposed a concept of fairness applicable to multiple protected groups in [6], which is defined as:

**Definition 1** *(Fairness) [6] Given a data matrix $\mathbf{X} \in \mathbb{R}^{d \times n}$ with $n$ instances and $d$ features, it is partitioned into $c$ disjoint clusters $\mathcal{C} = \{\pi_1, \cdots, \pi_c\}$. Given $t$ disjoint protected groups $\mathcal{G}_1, \mathcal{G}_2, \cdots, \mathcal{G}_t$, let $\eta_i = \frac{|\mathcal{G}_i|}{n}$ and $\eta_i(k) = \frac{|\pi_k \cap \mathcal{G}_i|}{|\pi_k|}$ denote the proportion of group $\mathcal{G}_i$ in the whole data and cluster $\pi_k$, respectively. The fairsess of a cluster $\pi_k$ is defined as:*

$$fairness\left(\pi_k\right) = \min\left(\frac{\eta_i}{\eta_i(k)}, \frac{\eta_i(k)}{\eta_i}\right), \ \forall i \in \{1, \cdots t\} \tag{5}$$

*The fairness of the whole clustering result $\mathcal{C}$ is defined as:*

$$fairness(\mathcal{C}) = \min_{k \in \{1, \cdots c\}} fairness(\pi_k) \tag{6}$$

**Remark 1** $fairness(\mathcal{C}) \in [0, 1]$, *and the larger $fairness(\mathcal{C})$ is, the fairer the clustering result is. A fair clustering result requires that the proportion of $\mathcal{G}_i$ in each cluster, which is denoted as $\eta_i(k)$, should be close to the proportion of $\mathcal{G}_i$ in the whole data, which is denoted as $\eta_i$. When all $\eta_i(k) = \eta_i$, the $fairness$ will achieve its maximum value 1, which means it is perfectly fair.*

Based on Definition 1, many fair clustering methods have been proposed [4, 7, 1, 37]. For example, Ziko et al. proposed a variational fair clustering framework by integrating fairness term with a clustering objective [57]; Kleindessner et al. embedded fairness as a linear constraint into spectral clustering obtaining fair spectral clustering [18]; Ghadiri et al. introduced a fair k-means method that ensures all protected groups have equal cluster costs [12]; Li et al. proposed a deep fair clustering method [20]. Wang et al. embedded this fairness into deep clustering by learning a differentiated and fair clustering allocation function [40]; Chhabra et al. provided a robust deep fair clustering method by considering the fairness attack [8].

# 3 Methodology

## 3.1 Fairness Regularization Term

We first introduce our fairness regularization term. To control the fairness, according to Definition 1, we need to compute $|\pi_k \cap \mathcal{G}_i|$ and $|\pi_k|$ in $\eta_i(k)$. To this end, we introduce two indicator matrices $\mathbf{G} \in \{0, 1\}^{n \times t}$ and $\mathbf{Y} \in \{0, 1\}^{n \times c}$. $\mathbf{G}$ is a protected group indicator matrix, where $G_{ij} = 1$ if the $i$-th instance belongs to the $j$-th protected group, and $G_{ij} = 0$ otherwise. $\mathbf{Y}$ is a cluster indicator matrix, where $Y_{ij} = 1$ if the $i$-th instance belongs to the $j$-th cluster, and $Y_{ij} = 0$ otherwise. It is easy to verify that

$$\mathbf{G}^T\mathbf{Y} = \begin{bmatrix} |\pi_1 \cap \mathcal{G}_1| & |\pi_2 \cap \mathcal{G}_1| & \dots & |\pi_c \cap \mathcal{G}_1| \\ |\pi_1 \cap \mathcal{G}_2| & |\pi_2 \cap \mathcal{G}_2| & \dots & |\pi_c \cap \mathcal{G}_2| \\ \vdots & \vdots & \ddots & \vdots \\ |\pi_1 \cap \mathcal{G}_t| & |\pi_2 \cap \mathcal{G}_t| & \dots & |\pi_c \cap \mathcal{G}_t| \end{bmatrix}, \ and \ \mathbf{Y}^T\mathbf{Y} = \begin{bmatrix} |\pi_1| & 0 & \dots & 0 \\ 0 & |\pi_2| & \dots & 0 \\ \vdots & \vdots & \ddots & \vdots \\ 0 & 0 & \dots & |\pi_c| \end{bmatrix} \tag{7}$$

Notice that $\mathbf{G}$ is a constant matrix because the protected groups are often pre-given, while $\mathbf{Y}$ is a variable that needs to learn for clustering. Based on Eq.(7), we define a fair regularization term $\text{Tr}\left(\mathbf{Y}^T\mathbf{G}\mathbf{G}^T\mathbf{Y}\left(\mathbf{Y}^T\mathbf{Y}\right)^{-1}\right)$ and provide the following Theorem, which shows that minimizing this regularization term leads to the maximum of the fairness defined in Definition 1.

**Theorem 1** *Given $\mathbf{G}$ and $\mathbf{Y}$ defined as mentioned before, we can obtain the maximum of fairness by optimizing the following objective function:*

$$\min_{\mathbf{Y}\in Ind} \text{Tr}\left(\mathbf{Y}^T\mathbf{G}\mathbf{G}^T\mathbf{Y}\left(\mathbf{Y}^T\mathbf{Y}\right)^{-1}\right). \tag{8}$$

**Proof 1** *We first have:*

$$\text{Tr}\left(\mathbf{Y}^T\mathbf{G}\mathbf{G}^T\mathbf{Y}\left(\mathbf{Y}^T\mathbf{Y}\right)^{-1}\right) = \text{Tr}\left(\left(\mathbf{Y}^T\mathbf{Y}\right)^{-\frac{1}{2}}\mathbf{Y}^T\mathbf{G}\mathbf{G}^T\mathbf{Y}\left(\mathbf{Y}^T\mathbf{Y}\right)^{-\frac{1}{2}}\right) = \left\|\mathbf{G}^T\mathbf{Y}\left(\mathbf{Y}^T\mathbf{Y}\right)^{-\frac{1}{2}}\right\|_F^2.$$

*According to Eq,(7), we have*

$$\mathbf{G}^T\mathbf{Y}\left(\mathbf{Y}^T\mathbf{Y}\right)^{-\frac{1}{2}} = \begin{bmatrix} \frac{|\pi_1\cap\mathcal{G}_1|}{\sqrt{|\pi_1|}} & \frac{|\pi_2\cap\mathcal{G}_1|}{\sqrt{|\pi_2|}} & \cdots & \frac{|\pi_c\cap\mathcal{G}_1|}{\sqrt{|\pi_c|}} \\ \frac{|\pi_1\cap\mathcal{G}_2|}{\sqrt{|\pi_1|}} & \frac{|\pi_2\cap\mathcal{G}_2|}{\sqrt{|\pi_2|}} & \cdots & \frac{|\pi_c\cap\mathcal{G}_2|}{\sqrt{|\pi_c|}} \\ \vdots & \vdots & \ddots & \vdots \\ \frac{|\pi_1\cap\mathcal{G}_t|}{\sqrt{|\pi_1|}} & \frac{|\pi_2\cap\mathcal{G}_t|}{\sqrt{|\pi_2|}} & \cdots & \frac{|\pi_c\cap\mathcal{G}_t|}{\sqrt{|\pi_c|}} \end{bmatrix} \tag{9}$$

*Therefore, minimizing Eq.(8) is equivalent to minimizing the following formula:*

$$\left\|\mathbf{G}^T\mathbf{Y}\left(\mathbf{Y}^T\mathbf{Y}\right)^{-\frac{1}{2}}\right\|_F^2 = \sum_{i=1}^t\sum_{k=1}^c \frac{|\pi_k\cap\mathcal{G}_i|^2}{|\pi_k|}. \tag{10}$$

*According to Cauchy-Schwarz Inequality, we have:*

$$\left(\sum_{k=1}^c \frac{|\pi_k\cap\mathcal{G}_i|^2}{|\pi_k|}\right)\left(\sum_{k=1}^c |\pi_k|\right) \geq \left(\sum_{k=1}^c |\pi_k\cap\mathcal{G}_i|\right)^2 = |\mathcal{G}_i|^2 \Rightarrow \sum_{k=1}^c \frac{|\pi_k\cap\mathcal{G}_i|^2}{|\pi_k|} \geq \frac{|\mathcal{G}_i|^2}{n}. \tag{11}$$

*Summing Eq.(11) w.r.t. $i$, we have*

$$\sum_{i=1}^t\sum_{k=1}^c \frac{|\pi_k\cap\mathcal{G}_i|^2}{|\pi_k|} \geq \sum_{i=1}^t \frac{|\mathcal{G}_i|^2}{n}. \tag{12}$$

*The equation in Eq.(12) holds if and only if $\frac{|\pi_1\cap\mathcal{G}_i|}{|\pi_1|} = \frac{|\pi_2\cap\mathcal{G}_i|}{|\pi_2|} = \cdots = \frac{|\pi_c\cap\mathcal{G}_i|}{|\pi_c|}$ for any $i$. It is easy to verify that $\frac{|\pi_1\cap\mathcal{G}_i|}{|\pi_1|} = \cdots = \frac{|\pi_c\cap\mathcal{G}_i|}{|\pi_c|} = \frac{\sum_k |\pi_k\cap\mathcal{G}_i|}{\sum_k |\pi_k|}$. Notice that $\pi_k$ is a disjoint partition of all data, and thus we have $(\pi_1\cap\mathcal{G}_i)\cup\cdots\cup(\pi_c\cap\mathcal{G}_i) = \mathcal{G}_i$ and $(\pi_p\cap\mathcal{G}_i)\cap(\pi_q\cap\mathcal{G}_i) = \emptyset$ for any $p,q$. Therefore, we have $\sum_k |\pi_k\cap\mathcal{G}_i| = |\mathcal{G}_i|$. Similarly, we have $\sum_k |\pi_k| = n$. Taking them back to the condition of the equation holding, we have that the equation holds if and only if $\frac{|\pi_1\cap\mathcal{G}_i|}{|\pi_1|} = \frac{|\pi_2\cap\mathcal{G}_i|}{|\pi_2|} = \cdots = \frac{|\pi_c\cap\mathcal{G}_i|}{|\pi_c|} = \frac{|\mathcal{G}_i|}{n}$.*

*Notice that $\frac{|\pi_k\cap\mathcal{G}_i|}{|\pi_k|} = \eta_i(k)$ and $\frac{|\mathcal{G}_i|}{n} = \eta_i$. Therefore, when we minimize Eq.(8), we have $\eta_i(k) = \eta_i$. According to Definition 1, it will lead to maximum fairness. This concludes the proof.*

According to Theorem 1, we provide a simple yet effective fair regularization term Eq.(8), and can easily plug it into the KKM and MKKM framework.

### 3.2 Fair Kernel K-means

Notice that the fairness regularization term $\text{Tr}\left(\mathbf{Y}^T\mathbf{G}\mathbf{G}^T\mathbf{Y}\left(\mathbf{Y}^T\mathbf{Y}\right)^{-1}\right)$ has a similar form to KKM (i.e., Eq.(2)). Therefore, we can seamlessly integrate this term into the KKM framework, leading to a fair kernel k-means (FKKM):

$$\min_{\mathbf{Y}\in Ind} \text{Tr}\left(\mathbf{K}\right) - \text{Tr}\left(\left(\mathbf{Y}^T\mathbf{Y}\right)^{-\frac{1}{2}}\mathbf{Y}^T\mathbf{K}\mathbf{Y}\left(\mathbf{Y}^T\mathbf{Y}\right)^{-\frac{1}{2}}\right) + \lambda\,\text{Tr}\left(\mathbf{Y}^T\mathbf{G}\mathbf{G}^T\mathbf{Y}\left(\mathbf{Y}^T\mathbf{Y}\right)^{-1}\right)$$

$$\iff \max_{\mathbf{Y}\in Ind} \text{Tr}\left(\mathbf{Y}^T\left(\mathbf{K} - \lambda\mathbf{G}\mathbf{G}^T\right)\mathbf{Y}\left(\mathbf{Y}^T\mathbf{Y}\right)^{-1}\right), \tag{13}$$

where $\lambda$ is a hyper-parameter to balance the trade-off between the clustering performance and the fairness. Larger $\lambda$ will lead to a fairer clustering result. Of course, $\lambda$ should not be too large, or it will dominate the loss function and the kernel k-means may not work. Comparing Eq.(13) with Eq.(2), we observe that if $\lambda$ is small enough to make $\mathbf{K} - \lambda\mathbf{G}\mathbf{G}^T$ positive semi-definite (p.s.d.), we can regard $\mathbf{K} - \lambda\mathbf{G}\mathbf{G}^T$ as a new kernel matrix and Eq.(13) becomes a standard kernel k-means. In this case, we call $\mathbf{K} - \lambda\mathbf{G}\mathbf{G}^T$ a *fair kernel*.

However, in practice, to make $\mathbf{K} - \lambda\mathbf{G}\mathbf{G}^T$ be a valid kernel matrix, which means to make $\mathbf{K} - \lambda\mathbf{G}\mathbf{G}^T$ p.s.d., we should set a very small $\lambda$, which cannot guarantee the fairness. To address this issue, we find that we can add a large enough constant term $\alpha\operatorname{Tr}(\mathbf{I})$ to Eq.(13), to obtain a valid fair kernel matrix. In more detail, we have:

$$\operatorname{Tr}\left(\mathbf{Y}^T\left(\mathbf{K} - \lambda\mathbf{G}\mathbf{G}^T\right)\mathbf{Y}\left(\mathbf{Y}^T\mathbf{Y}\right)^{-1}\right) + \alpha\operatorname{Tr}(\mathbf{I}) = \operatorname{Tr}\left(\mathbf{Y}^T\left(\mathbf{K} + \alpha\mathbf{I} - \lambda\mathbf{G}\mathbf{G}^T\right)\mathbf{Y}\left(\mathbf{Y}^T\mathbf{Y}\right)^{-1}\right). \quad (14)$$

It shows that optimizing Eq.(14) is always exactly equivalent to optimizing Eq.(13), no matter how we set $\alpha$. With a large enough $\alpha$, we can easily set an appropriate $\lambda$ to make $\tilde{\mathbf{K}} = \mathbf{K} + \alpha\mathbf{I} - \lambda\mathbf{G}\mathbf{G}^T$ be p.s.d., and thus be a valid kernel matrix. We will discuss how to set $\lambda$ and $\alpha$ later.

In this way, we obtain an extremely simple yet elegant FKKM method. In this method, we do not even need to modify the loss of standard KKM. All we need is to modify the kernel by replacing $\mathbf{K}$ to a fair kernel $\tilde{\mathbf{K}} = \mathbf{K} + \alpha\mathbf{I} - \lambda\mathbf{G}\mathbf{G}^T$. It means that we realize the fairness on the data level rather than the model level.

### 3.3 Fair Multiple Kernel K-means

Eq.(14) can be naturally extended to a multiple kernel setting. Given a base kernel $\mathbf{K}^{(p)}$, we first construct its fair kernel $\tilde{\mathbf{K}}^{(p)} = \mathbf{K}^{(p)} + \alpha\mathbf{I} - \lambda\mathbf{G}\mathbf{G}^T$. Then similar to Eq.(4), we define the fair consensus kernel $\tilde{\mathbf{K}}^* = \sum_{p=1}^m \gamma_p^2\tilde{\mathbf{K}}^{(p)}$ and take it into Eq.(2) to obtain FMKKM:

$$\min_{\mathbf{Y},\boldsymbol{\gamma}}\operatorname{Tr}\left(\tilde{\mathbf{K}}^*\left(\mathbf{I} - \mathbf{Y}\left(\mathbf{Y}^T\mathbf{Y}\right)^{-1}\mathbf{Y}^T\right)\right) \quad s.t.\ \mathbf{Y} \in Ind,\ \boldsymbol{\gamma}^T\mathbf{1} = 1,\ \gamma_p \geq 0,\ \tilde{\mathbf{K}}^* = \sum_{p=1}^m \gamma_p^2\tilde{\mathbf{K}}^{(p)}. \quad (15)$$

Notice that since our fairness regularization term $\operatorname{Tr}\left(\mathbf{Y}^T\mathbf{G}\mathbf{G}^T\mathbf{Y}\left(\mathbf{Y}^T\mathbf{Y}\right)^{-1}\right)$ requires that $\mathbf{Y}$ should be a discrete indicator matrix, our FKKM (i.e., Eq.(14)) and FMKKM (i.e., Eq.(15)) directly solve the discrete $\mathbf{Y}$ instead of the conventional two-step methods which learn an orthogonal embedding $\mathbf{H}$ first and then obtain the discrete clustering result. As we know, in the two-step methods, the kernel k-means and the discretization post-processing are separated and when doing the discretization it cannot guarantee the clustering accuracy or fairness. Different from the two-step methods, we can directly learn the final clustering result $\mathbf{Y}$ by fully considering the clustering accuracy and fairness.

### 3.4 Optimization

#### 3.4.1 Optimization of FKKM

When minimizing Eq.(14), we only need to solve one variable $\mathbf{Y}$. Notice that there is only one 1 in each row of $\mathbf{Y}$. Therefore, we can solve $\mathbf{Y}$ row by row. When solving the $i$-th row, we replace the $i$-th row with $[1, 0, \cdots, 0]$, $[0, 1, 0, \cdots, 0]$, ..., $[0, \cdots, 0, 1]$ respectively, and compute the values of the corresponding objective function to find the one which leads to the maximum. Then we set the $i$-th row as this row vector. Wang et al. propose an efficient method to compute these objective functions by reducing the computation redundancy [43].

#### 3.4.2 Optimization of FMKKM

In Eq.(15), there are two groups of variables, i.e., $\mathbf{Y}$ and $\boldsymbol{\gamma}$. We solve them by a block coordinate descent method, which optimizes one variable when fixing the other.

When fixing $\boldsymbol{\gamma}$ to solve $\mathbf{Y}$, we have the following subproblem w.r.t $\mathbf{Y}$:

$$\max_{\mathbf{Y} \in Ind}\operatorname{Tr}\left(\mathbf{Y}^T\tilde{\mathbf{K}}^*\mathbf{Y}\left(\mathbf{Y}^T\mathbf{Y}\right)^{-1}\right), \quad (16)$$

where $\tilde{\mathbf{K}}^* = \sum_{p=1}^m \gamma_p^2\tilde{\mathbf{K}}^{(p)}$. It is the same as the optimization of FKKM.

When fixing $\mathbf{Y}$ to solve $\boldsymbol{\gamma}$, we have following subproblem w.r.t $\boldsymbol{\gamma}$:

$$\min_{\boldsymbol{\gamma}} \sum_{p=1}^{m} \gamma_p^2 h_p, \ \ s.t. \sum_{p=1}^{m} \gamma_p = 1, \ \gamma_p \geq 0, \tag{17}$$

where $h_p = \mathrm{Tr}\left(\tilde{\mathbf{K}}^{(p)} \left(\mathbf{I} - \mathbf{Y}\left(\mathbf{Y}^T\mathbf{Y}\right)^{-1}\mathbf{Y}^T\right)\right)$. According to Cauchy-Schwarz Inequality, the closed-form solution of $\gamma_p$ is:

$$\gamma_p = \frac{h_p^{-1}}{\sum_{j=1}^{m} h_j^{-1}}. \tag{18}$$

Appendix A shows the pseudo-codes of FKKM and FMKKM, respectively. When updating each row of $\mathbf{Y}$, the objective function of FKKM decreases and has a lower bound. Therefore, FKKM can always converge. Similarly, the convergence of FMKKM can also be guaranteed. Now, we analyze the time complexity. According to [43], optimizing the $i$-th row of $\mathbf{Y}$ has a time complexity of $O(nc)$. FKKM has a time complexity of $O(n^2c)$. Calculating $\boldsymbol{\gamma}$ has a time complexity of $O(n)$. Therefore, FMKKM also has a time complexity of $O(n^2c)$. According to [43], although the time complexity is square in the number of instances, it can be computed very efficiently in practice. Therefore, the time complexity of our method is comparable with the mainstream KKM and MKKM methods.

## 4 Theoretical Analysis

The generalization error bound of the k-means evaluates the expectation of distance between an unseen data and the clustering center it belongs to [30, 22, 21]. Since FKKM is a special case of FMKKM when $m = 1$, in this section, we derive the generalization error bound of our FMKKM. Before the derivation, we need the following two mild assumptions:

**Assumption 1** *Each* $\tilde{\mathbf{K}}^{(p)} = \mathbf{K}^{(p)} + \alpha\mathbf{I} - \lambda\mathbf{G}\mathbf{G}^T$ *is a valid kernel matrix, i.e.,* $\tilde{\mathbf{K}}^{(p)}$ *is symmetric and p.s.d.*

**Remark 2** *This assumption is easy to satisfy. If* $\tilde{\mathbf{K}}^{(p)}$ *is not p.s.d., we can enlarge* $\alpha$ *to make the assumption hold.*

**Assumption 2** *All* $\mathbf{K}^{(p)}$ *are upper bounded. We denote* $b$ *as the maximum of elements in all* $\mathbf{K}^{(p)}$.

According to assumption 1, since all $\tilde{\mathbf{K}}^{(p)}$ are valid kernel matrices, $\tilde{\mathbf{K}}^*$ is also a valid kernel matrix. We define the corresponding kernel function of $\tilde{\mathbf{K}}^*$ as $\tilde{\mathcal{K}}^*(\cdot, \cdot)$, and its kernel mapping function is $\Phi_{\boldsymbol{\gamma}}(\mathbf{x}_i) = [\gamma_1\Phi_1(\mathbf{x}_i)^T, \ldots, \gamma_m\Phi_m(\mathbf{x}_i)^T]^T : \mathbb{R}^d \mapsto \mathcal{H}$, where $\Phi_1(\mathbf{x}_i), \ldots, \Phi_m(\mathbf{x}_i)$ are the induced kernel mapping function of $\tilde{\mathbf{K}}^{(1)}, \ldots, \tilde{\mathbf{K}}^{(m)}$, respectively. Let $\mathbf{M} = [\mathbf{m}_1, \ldots, \mathbf{m}_c]$ denote the learned centroids matrix in the RKHS $\mathcal{H}$, where $\mathbf{m}_i$ is the center of the $i$-th cluster in $\mathcal{H}$. FMKKM aims to minimize the error: $\mathbb{E}\left[\min_{\mathbf{y} \in \{\mathbf{e}_1, \ldots, \mathbf{e}_c\}} \|\Phi_{\boldsymbol{\gamma}}(\mathbf{x}) - \mathbf{M}\mathbf{y}\|_{\mathcal{H}}^2\right]$, where $[\mathbf{e}_1, \ldots, \mathbf{e}_c]$ are the standard orthonormal basis of $\mathbb{R}^c$ space, i.e., $\mathbf{e}_i$ is an all-zero vectors except that the $i$-th element is 1.

Then, we define a function class as our hypothesis space:

$$\mathcal{F} = \left\{f : \mathbf{x} \mapsto \min_{\mathbf{y} \in \{\mathbf{e}_1, \ldots, \mathbf{e}_c\}} \|\Phi_{\boldsymbol{\gamma}}(\mathbf{x}) - \mathbf{M}\mathbf{y}\|_{\mathcal{H}}^2 \,\middle|\, \boldsymbol{\gamma}^T\mathbf{1} = 1, \gamma_p \geq 0, \mathbf{m}_k \in \mathcal{H}\right\}. \tag{19}$$

Similar to [21], we have the following Theorem to provide the generalization error bound:

**Theorem 2** *Under Assumptions 1 and 2, given training data* $\mathbf{X} = [\mathbf{x}_1, \ldots, \mathbf{x}_n]$, *function class* $\mathcal{F}$ *defined in Eq.(19), and any* $\delta \geq 0$, *with probability at least* $1 - \delta$, *the following inequality holds for all* $f \in \mathcal{F}$:

$$\mathbb{E}[f(\mathbf{x})] \leq \frac{1}{n}\sum_{i=1}^{n} f(\mathbf{x}_i) + \frac{2\sqrt{2\pi}}{\sqrt{n}}\left[(1 + c^2)(b + \alpha) - (1 + \frac{c^2}{t})\lambda + c\sqrt{2(b + \alpha - \lambda)\left(b + \alpha - \frac{\lambda}{t}\right)}\right]$$

$$+ \left(4(b + \alpha) - 2\left(1 + \frac{1}{t}\right)\lambda\right)\sqrt{\frac{\log(1/\delta)}{2n}}, \tag{20}$$

*where* $t$ *and* $c$ *are the number of protected groups and clusters, respectively.*

**Proof 2** *See Appendix B.*

The first term in Eq.(20) is the empirical error. Notice that, we have $\sum_{i=1}^{n} f(\mathbf{x}_i) = \mathrm{Tr}\left(\tilde{\mathbf{K}}^* \left(\mathbf{I} - \mathbf{Y}\left(\mathbf{Y}^T\mathbf{Y}\right)^{-1}\mathbf{Y}^T\right)\right)$, which means our loss function is to minimize exactly this empirical error. However, in the two-step methods, which apply $\mathbf{H}^T\mathbf{H} = \mathbf{I}$ where $\mathbf{H} = \mathbf{Y}\left(\mathbf{Y}^T\mathbf{Y}\right)^{-\frac{1}{2}}$ to replace $\mathbf{Y} \in Ind$, they only optimize a continual approximation of the empirical error.

Besides, the second and third terms represent the gap between the generalization and empirical errors. Intuitively, the gap is the smaller the better. To decrease the gap, we wish $\alpha$ to be as small as possible. However, Assumption 1 prevents $\alpha$ being too small because $\tilde{\mathbf{K}}^{(p)}$ should be p.s.d., or Theorem 2 will not hold anymore. Now we can derive the lower bound of $\alpha$ according to Assumption 1. Suppose $\sigma_{min}$ as the smallest eigenvalue of $\mathbf{K}^{(1)}, \ldots, \mathbf{K}^{(p)}$. Then, the smallest eigenvalue of $\mathbf{K}^{(p)} + \alpha\mathbf{I}$ should be no smaller than $\sigma_{min} + \alpha$. Notice that we have the following Lemma:

**Lemma 1** *Given two real symmetric matrices $\mathbf{A}$ and $\mathbf{B}$ with the same size, where the smallest eigenvalue of $\mathbf{A}$ is $\sigma_A$ and the largest eigenvalue of $\mathbf{B}$ is $\sigma_B$. If $\sigma_A \geq \sigma_B$, then $\mathbf{A} - \mathbf{B}$ is p.s.d.*

**Proof 3** *See Appendix C.*

Denoting $\sigma_{max}$ as the largest eigenvalue of $\mathbf{G}\mathbf{G}^T$, it is easy to verify that $\sigma_{max} = |\mathcal{G}_{max}|$, where $\mathcal{G}_{max}$ is the protected group with the largest number of instances. According to Lemma 1, we have that if $\sigma_{min} + \alpha - \lambda * |\mathcal{G}_{max}| \geq 0$, $\tilde{\mathbf{K}}^{(p)}$ will be p.s.d. Therefore, $\alpha$ has a lower bound $\lambda * |\mathcal{G}_{max}| - \sigma_{min}$. In practice, $\sigma_{min}$ is often very small and close to 0. To avoid the time consuming to compute the eigenvalues of the kernels, we can approximately set $\alpha = \lambda * |\mathcal{G}_{max}|$.

Take $\alpha = \lambda * |\mathcal{G}_{max}|$ back into the generalization error bound Eq.(20). We consider the gap between the generalization and empirical errors, i.e., the second and third terms:

$$\frac{2\sqrt{2\pi}}{\sqrt{n}}\left[(1+c^2)(b+\alpha) - (1+\frac{c^2}{t})\lambda + c\sqrt{2(b+\alpha-\lambda)\left(b+\alpha-\frac{\lambda}{t}\right)}\right] + \left(4(b+\alpha) - 2\left(1+\frac{1}{t}\right)\lambda\right)\sqrt{\frac{\log(1/\delta)}{2n}}$$

$$\geq \frac{2\sqrt{2\pi}}{\sqrt{n}}\left[(1+c^2)b + \left(|\mathcal{G}_{max}| - 1 + \frac{c^2(|\mathcal{G}_{max}|t-1)}{t}\right)\lambda + c\sqrt{2(b+(|\mathcal{G}_{max}|-1)\lambda)\left(b+\frac{|\mathcal{G}_{max}|t-1}{t}\lambda\right)}\right]$$

$$+ (4b + 4(|\mathcal{G}_{max}|-1)\lambda)\sqrt{\frac{\log(1/\delta)}{2n}} \tag{21}$$

Notice that $|\mathcal{G}_{max}| - 1 \geq 0$ and $|\mathcal{G}_{max}|t - 1 \geq 0$, and thus we have that the gap decreases with $\lambda$ decreases. It means that smaller $\lambda$ leads to a lower gap. Therefore, $\lambda$ is a trade-off between the clustering performance and fairness. Increasing $\lambda$ may enlarge the error bound, but obtain a fairer result. Based on this theoretical analysis, we provide a strategy to set $\lambda$ by observing a fairness metric, which can be computed without the ground truth. In more detail, we gradually enlarge $\lambda$ from 0, set $\alpha = \lambda * |\mathcal{G}_{max}|$, and observe the fairness metric. If it gets stable good fairness, we stop enlarging $\lambda$ and set $\lambda$ as the current value. This strategy does not need the ground truth, which is appropriate for unsupervised learning, and can obtain an as small as possible $\lambda$ to achieve a good fairness result.

## 5 Experiments

### 5.1 Data Sets and Experimental Setup

We conduct experiments on benchmark data sets which are widely used in fair clustering, including D&S [2], HAR [3], Jaffe [29], MNIST-USPS [20], Credit Card [52] and K1b [53]. D&S is a human daily and sports activities data set including 8 participants. HAR is a human action recognition data set including 30 participants. In both D&S and HAR data sets, the data of each participant form a protected group. Jaffe is a face image data set. Following [20], the face images with the same expressions are put into a protected group. MNIST-USPS is an image data set containing images of handwritten digits from the subsets of MNIST and USPS data sets. Following [20], we randomly sample 2000 images from MNIST to form one protected group and randomly sample 1800 images from USPS to form the other protected group. Credit card is a data set that describes the customers' default payments and the data of males and females form two protected groups respectively. K1b is a

text data set. Following [48], we randomly assign each text to a protected group with a Bernoulli distribution whose $p = 0.5$ to form two protected groups. The statistical information of these data sets is shown in Appendix D.

Table 1: Comparison results on the single kernel setting. The best and second best results are denoted in **bold** and underlined, respectively.

| Data sets | | K-means | KKM | SC | FairSC | VFC | FFC | FKKM-f | FKKM |
|---|---|---|---|---|---|---|---|---|---|
| D&S | ACC | 0.555 | 0.552 | 0.558 | 0.433 | 0.539 | 0.521 | **0.648** | 0.636 |
| | NMI | 0.650 | 0.602 | 0.652 | 0.575 | 0.617 | 0.583 | **0.724** | 0.683 |
| | Bal | 0 | 0 | 0 | 0 | 0.186 | 0.100 | 0 | **0.559** |
| | MNCE | 0.156 | 0.531 | 0.023 | 0 | 0.923 | 0.712 | 0.477 | **0.991** |
| HAR | ACC | 0.524 | 0.620 | 0.680 | 0.742 | 0.600 | 0.602 | 0.689 | **0.771** |
| | NMI | 0.596 | 0.609 | 0.618 | 0.703 | 0.654 | 0.490 | 0.625 | **0.710** |
| | Bal | 0 | 0 | 0 | 0 | 0.200 | 0.007 | 0 | **0.250** |
| | MNCE | 0.933 | 0.930 | 0.914 | 0 | 0.983 | 0.953 | 0.920 | **0.989** |
| MNIST-USPS | ACC | 0.363 | 0.396 | 0.406 | **0.458** | 0.360 | 0.437 | 0.403 | 0.432 |
| | NMI | 0.423 | 0.421 | **0.435** | 0.429 | 0.306 | 0.412 | 0.426 | 0.380 |
| | Bal | 0 | 0 | 0 | 0 | 0.142 | 0.217 | 0 | **0.847** |
| | MNCE | 0 | 0.003 | 0 | 0 | 0.544 | 0.684 | 0 | **0.997** |
| Jaffe | ACC | 0.927 | 0.948 | 0.901 | 0.957 | 0.981 | 0.901 | 0.954 | **1** |
| | NMI | 0.914 | 0.922 | 0.889 | 0.943 | 0.969 | 0.918 | 0.930 | **1** |
| | Bal | 0 | 0 | 0 | 0 | 0.400 | 0.250 | 0 | **0.500** |
| | MNCE | 0.808 | 0.900 | 0.765 | 0.827 | 0.983 | 0.924 | 0.897 | **0.989** |
| Credit Card | ACC | 0.362 | 0.381 | 0.311 | 0.351 | 0.381 | 0.364 | 0.400 | **0.404** |
| | NMI | 0.139 | 0.140 | 0.126 | 0.123 | 0.142 | 0.139 | 0.145 | **0.148** |
| | Bal | 0.510 | 0.550 | 0.567 | 0.603 | 0.586 | 0.550 | 0.536 | **0.624** |
| | MNCE | 0.953 | 0.961 | 0.967 | 0.973 | 0.970 | 0.969 | 0.956 | **0.985** |
| K1b | ACC | 0.742 | 0.669 | 0.667 | **0.853** | 0.778 | 0.663 | 0.826 | 0.809 |
| | NMI | 0.589 | 0.537 | 0.536 | **0.666** | 0.553 | 0.503 | 0.628 | 0.591 |
| | Bal | 0.666 | 0.775 | 0.763 | 0.667 | 0.794 | 0.773 | 0.703 | **0.800** |
| | MNCE | 0.971 | 0.989 | 0.987 | 0.971 | 0.990 | 0.989 | 0.978 | **0.991** |

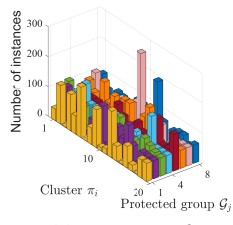

(a) FMKKM-f

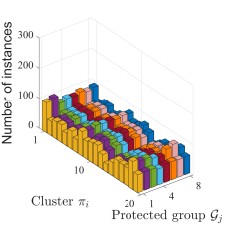

(b) FMKKM

Figure 1: Fairness visualization results of FMKKM-f and FMKKM on D&S.

In the single kernel setting, we compare our FKKM with K-means [13], Kernel K-means (KKM) [10], Spectral Clustering (SC) [33], and three state-of-the-art fair clustering methods, including SpFC [18], VFC [58], and FFC [34]. For the kernel methods (i.e., our FKKM and KKM), we use a Gaussian kernel with a bandwidth parameter fixing to $\sqrt{0.5} * D$, where $D$ is the average distance between samples. In the multiple kernel setting, we compare our FMKKM with 9 state-of-the-art MKKM methods, including ONKC [25], MKCSS [55], DPMKKM [42], LFLKA [51], EMKC [38], OSLR [47], ASLR [46], CSAMKC [56], FAMKKM [41]. Detailed information of these compared methods is shown in Appendix E. Besides, for an ablation study, we also compare with the degeneration version of our method, which is without the fairness regularization term, denoted as FKKM-f (for single kernel version) and FMKKM-f (for multiple kernel version).

In the multiple kernel setting, following [11], we construct 12 kernels, including seven Gaussian kernels $\mathbf{K}(\mathbf{x}_i, \mathbf{x}_j) = \exp\left(-\|\mathbf{x}_i - \mathbf{x}_j\|_2^2/2\epsilon^2\right)$ with $\epsilon = \sqrt{s} * D$, where $s$ varies in the range of $\{\frac{1}{8}, \frac{1}{4}, \frac{1}{2}, 1, 2, 4, 8\}$ and $D$ is the average distance between samples; four polynomial kernels $\mathbf{K}(\mathbf{x}_i, \mathbf{x}_j) = \left(a + \mathbf{x}_i^T \mathbf{x}_j\right)^b$ with $a = \{0, 1\}$ and $b = \{2, 4\}$; and a cosine kernel $\mathbf{K}(\mathbf{x}_i, \mathbf{x}_j) = \left(\mathbf{x}_i^T \mathbf{x}_j\right) / (\|\mathbf{x}_i\| \cdot \|\mathbf{x}_j\|)$. Finally, all kernels have been normalized through $\mathbf{K}(\mathbf{x}_i, \mathbf{x}_j)/\sqrt{\mathbf{K}(\mathbf{x}_i, \mathbf{x}_i)\mathbf{K}(\mathbf{x}_j, \mathbf{x}_j)}$ and then rescaled to $[0, 1]$. We use Accuracy (ACC) and Normalized Mutual Information (NMI) to evaluate the clustering performance. Besides, we also use balance (Bal) [20] and Minimal Normalized Conditional Entropy (MNCE) [49] to evaluate fairness. Specifically, Bal is defined as

$$\text{Bal}(\mathcal{C}) = \min_k \left( \frac{N_k^{\min}}{N_k^{\max}} \right) \in [0, 1], \quad (22)$$

where $N_k^{min}$ and $N_k^{max}$ represent the number of instances in the smallest and the largest (in size) protected groups in cluster $\pi_k$, respectively. MNCE is defined as

$$\text{MNCE} = \frac{\min_k \left( -\sum_i \frac{|\mathcal{G}_i \cap \pi_k|}{|\pi_k|} \log \frac{|\mathcal{G}_i \cap \pi_k|}{|\pi_k|} \right)}{-\sum_i \frac{|\mathcal{G}_i|}{n} \log \frac{|\mathcal{G}_i|}{n}} \in [0, 1]. \quad (23)$$

All metrics are the larger the better. Based on previous analysis of hyper-parameter setting, we search $\lambda$ as $\lambda = 1, 2, \ldots$, by observing the corresponding MNCE. When the MNCE gets stable, i.e., the change of MNCE is smaller than 0.005, we stop the searching and use the current $\lambda$. For

Table 2: Comparison results on the multiple kernel setting. The best and second best results are denoted in **bold** and underlined, respectively.

| Data sets | | ONKC | MKCSS | DPMKKM | LFLKA | EMKC | OSLR | ASLR | CSAMKC | FAMKKM | FMKKM-f | FMKKM |
|---|---|---|---|---|---|---|---|---|---|---|---|---|
| D&S | ACC | 0.505 | 0.543 | 0.614 | **0.646** | 0.491 | 0.590 | 0.508 | 0.598 | 0.601 | 0.645 | 0.616 |
| | NMI | 0.644 | 0.665 | 0.697 | 0.717 | 0.602 | 0.677 | 0.591 | 0.668 | 0.678 | **0.718** | 0.661 |
| | Bal | 0 | 0 | 0 | 0 | 0 | 0 | 0 | 0 | 0 | 0 | **0.471** |
| | MNCE | 0.333 | 0 | 0.333 | 0.622 | 0.501 | 0.649 | 0 | 0.598 | 0.476 | 0.585 | **0.985** |
| HAR | ACC | 0.526 | 0.646 | 0.692 | 0.695 | 0.732 | 0.717 | 0.574 | 0.668 | 0.705 | 0.697 | **0.791** |
| | NMI | 0.557 | 0.670 | 0.622 | 0.622 | 0.656 | 0.650 | 0.549 | 0.558 | 0.642 | 0.655 | **0.752** |
| | Bal | 0 | 0 | 0 | 0 | 0 | 0 | 0 | 0 | 0 | 0 | **0.263** |
| | MNCE | 0.933 | 0.920 | 0.905 | 0.914 | 0.939 | 0.917 | 0.520 | 0.928 | 0.923 | 0.888 | **0.990** |
| MNIST-USPS | ACC | 0.397 | 0.457 | 0.391 | 0.412 | 0.415 | 0.406 | 0.436 | 0.398 | 0.445 | 0.412 | **0.495** |
| | NMI | 0.400 | 0.442 | 0.359 | 0.407 | 0.406 | 0.406 | 0.449 | 0.382 | 0.402 | 0.416 | **0.454** |
| | Bal | 0 | 0 | 0 | 0 | 0 | 0 | 0 | 0.024 | 0 | 0 | **0.808** |
| | MNCE | 0 | 0 | 0 | 0 | 0 | 0 | 0 | 0.161 | 0 | 0 | **0.993** |
| Jaffe | ACC | 0.840 | 0.956 | 0.939 | 0.911 | 0.967 | 0.934 | 0.921 | 0.948 | 0.985 | 0.939 | **0.995** |
| | NMI | 0.848 | 0.958 | 0.924 | 0.887 | 0.964 | 0.903 | 0.936 | 0.925 | 0.971 | 0.914 | **0.991** |
| | Bal | 0 | 0.200 | 0 | 0 | 0 | 0 | 0 | 0 | 0.250 | 0 | **0.500** |
| | MNCE | 0.542 | 0.880 | 0 | 0.826 | 0.964 | 0.923 | 0.686 | 0.917 | 0.970 | 0.895 | **0.989** |
| Credit Card | ACC | **0.402** | 0.333 | 0.363 | 0.360 | 0.337 | 0.370 | 0.321 | 0.327 | 0.355 | 0.378 | 0.375 |
| | NMI | 0.141 | 0.139 | 0.126 | 0.135 | 0.119 | 0.138 | 0.103 | 0.091 | 0.123 | **0.148** | 0.147 |
| | Bal | 0.547 | 0.558 | 0.523 | 0.590 | 0.557 | 0.599 | 0.587 | 0.497 | 0.571 | 0.559 | **0.641** |
| | MNCE | 0.960 | 0.964 | 0.950 | 0.975 | 0.963 | 0.977 | 0.973 | 0.938 | 0.969 | 0.964 | **0.989** |
| K1b | ACC | 0.692 | 0.688 | 0.723 | 0.687 | 0.601 | 0.623 | **0.850** | 0.749 | 0.745 | 0.826 | 0.828 |
| | NMI | 0.435 | 0.535 | 0.286 | 0.545 | 0.436 | 0.523 | **0.652** | 0.554 | 0.581 | 0.632 | 0.601 |
| | Bal | 0.428 | 0.794 | 0.545 | 0.818 | 0.881 | 0.834 | 0.714 | 0.892 | 0.849 | 0.757 | **0.935** |
| | MNCE | 0.881 | 0.991 | 0.937 | 0.993 | 0.997 | 0.994 | 0.980 | 0.998 | 0.995 | 0.986 | **1** |

other comparison methods, we follow their recommended parameter configurations and search methodologies. All experiments are conducted on the 12th Gen Interl(R) Core(TM) i7-12700 with 32 GB RAM. All experiments are repeated 10 times and the average results are reported.

## 5.2 Experimental Results

Table 1 shows the comparison results in the single kernel setting, where the best and second best results are denoted in bold and underlined, respectively. It can be seen that FKKM exhibits better fairness compared to K-means, KKM, SC, our ablation version (i.e., FKKM-f), and even the fair clustering methods, indicating the effectiveness of our fairness regularization term. When comparing w.r.t. clustering performance (i.e., ACC and NMI), FKKM still often achieves the best or the second-best results.

Table 2 presents the comparison results in the multiple kernel setting. FMKKM easily achieves the best fairness, due to the effectiveness of our fairness regularization term. Moreover, FMKKM often achieves better or at least comparable ACC and NMI. Notice that our method just simply modifies the original MKKM and can achieve competitive clustering performance, demonstrating that our method is simple yet effective.

Figure 1 shows the visualization results. It shows the number of instances of each protected group $\mathcal{G}_j$ in each cluster $\pi_i$ in the D&S data set obtained by FMKKM-f and FMKKM, respectively. As shown in Figure 1(a), in FMKKM-f, without the fairness regularization term, the numbers of data of each protected group in each cluster have a great difference, which means the result is unfair. Figure 1 (b) shows that the distribution of protected group in each cluster is more balanced, which means the result obtained by FMKKM is much fairer than FMKKM-f. It demonstrates the effectiveness of our fair regularization term.

## 5.3 Efficiency Results

The convergence curves of our methods are shown in Appendix F. The results show that our methods often converge very fast. We also conduct experiments to compare the running time of our methods with other compared methods. Our method is faster than or at least comparable with other methods on many data sets. The detailed results are shown in Appendix F.

## 5.4 Parameter Study

Figure 2 shows the effects of $\lambda$ of FKKM and FMKKM on MNIST-USPS and Credit Cards data sets. The other results are similar. The red points denote the $\lambda$ selected by our strategy. We can see that

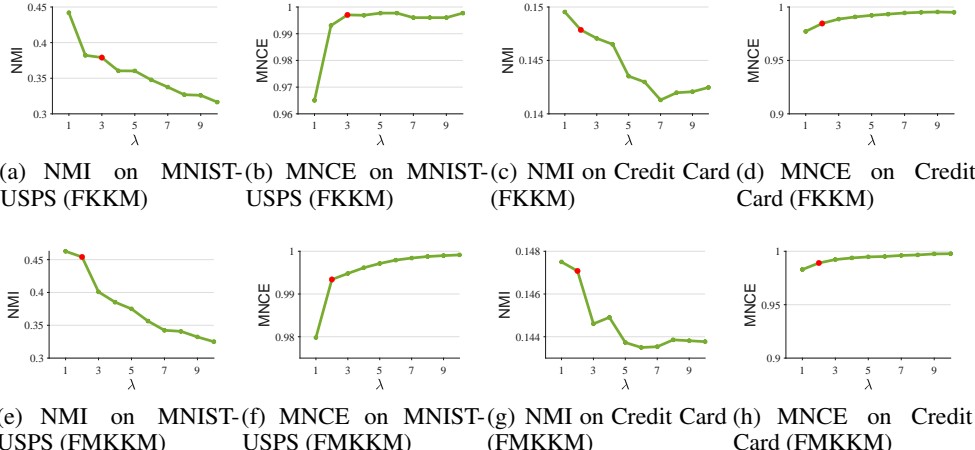

(a) NMI on MNIST-USPS (FKKM)  (b) MNCE on MNIST-USPS (FKKM)  (c) NMI on Credit Card (FKKM)  (d) MNCE on Credit Card (FKKM)

(e) NMI on MNIST-USPS (FMKKM)  (f) MNCE on MNIST-USPS (FMKKM)  (g) NMI on Credit Card (FMKKM)  (h) MNCE on Credit Card (FMKKM)

Figure 2: NMI and MNCE of our methods on MNIST-USPS and Credit Card data sets w.r.t. different values of $\lambda$. The red points represent the lambda that our algorithm automatically searches for.

with the increase of $\lambda$, the fairness grows and the clustering performance may decrease, which is consistent with our previous discussion. We can often achieve a good trade-off between fairness and performance at the red point, which shows the effectiveness of our hyper-parameter setting strategy.

## 6  Conclusion

In this paper, we focused on the fairness issue in KKM and MKKM. We carefully designed a novel fairness regularization term, which can be seamlessly plugged into the KKM and MKKM framework. Equipped with this fairness regularization term, we proposed a novel FKKM and FMKKM method. We also provided a hyper-parameter setting strategy based on the theoretical analysis to make the methods easy to use. Extensive experiments demonstrated the effectiveness and superiority of our proposed FKKM and FMKKM methods.

Although the proposed methods achieve promising performance on fairness, they still have some limitations. For example, in our methods, the protected groups must be pre-given or decided by humans. An interesting question is how to automatically decide the protected groups without human intervention. In the future, we will focus on this problem.

## Acknowledgments

This work is supported by the National Natural Science Foundation of China grants 62176001 and 62376146, and Natural Science Project of Anhui Provincial Education Department grants 2023AH030004.

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

# A Pseudo-codes of FKKM and FMKKM

Algorithms 1 and 2 show the pseudo-codes of our FKKM and FMKKM, respectively.

---

**Algorithm 1** Fair Kernel K-means

---

**Input:** Kernel matrix $\mathbf{K}$, protected groups $\mathcal{G}_1, \cdots, \mathcal{G}_t$, fairness hyper-parameter $\lambda$.
 1: Construct protected group indicator matrix $\mathbf{G}$ and calculate $\alpha$ as $\alpha = |\mathcal{G}_{max}| * \lambda$.
 2: Construct the fair kernel by $\tilde{\mathbf{K}} = \mathbf{K} + \alpha\mathbf{I} - \lambda\mathbf{GG}^T$.
 3: Initialize $\mathbf{Y}$ by running standard kernel k-means on $\mathbf{K}$.
 4: **repeat**
 5:    Update $\mathbf{Y}$ row by row by maximize $\text{Tr}\left(\mathbf{Y}^T\tilde{\mathbf{K}}\mathbf{Y}\left(\mathbf{Y}^T\mathbf{Y}\right)^{-1}\right)$.
 6: **until** Converges
**Output:** The final partition matrix $\mathbf{Y}$.

---

**Algorithm 2** Fair Multiple Kernel K-means

---

**Input:** Kernel matrices $\{\mathbf{K}^{(p)}\}_{p=1}^m$, protected groups $\mathcal{G}_1, \cdots, \mathcal{G}_t$, fairness hyper-parameter $\lambda$.
 1: Construct protected group indicator matrix $\mathbf{G}$ and calculate $\alpha$ as $\alpha = |\mathcal{G}_{max}| * \lambda$.
 2: Construct the corresponding fair kernel by $\tilde{\mathbf{K}}^{(p)} = \mathbf{K}^{(p)} + \alpha\mathbf{I} - \lambda\mathbf{GG}^T$ for each base kernel matrix $\mathbf{K}^{(p)}$.
 3: Initialize $\boldsymbol{\gamma} = \frac{1}{m}$ and $\mathbf{Y}$ by running standard kernel k-means on $\sum_{p=1}^m \gamma_p^2 \mathbf{K}^{(p)}$.
 4: **repeat**
 5:    Update $\mathbf{Y}$ row by row by solving Eq.(16).
 6:    Update $\boldsymbol{\gamma}$ by Eq.(18)
 7: **until** Converges
**Output:** The final partition matrix $\mathbf{Y}$.

---

# B Proof of Theorem 2

Denote $\hat{R}(\mathbf{M}, \boldsymbol{\gamma}) = \frac{1}{n}\sum_{i=1}^n \min_{\mathbf{y}\in\{\mathbf{e}_1,\dots,\mathbf{e}_c\}} \|\Phi_{\boldsymbol{\gamma}}(\mathbf{x}_i) - \mathbf{My}\|_{\mathcal{H}}^2$. Our goal is to bound:

$$\sup_{f\in\mathcal{F}} \left(\mathbb{E}[f(\mathbf{x})] - \frac{1}{n}\sum_{i=1}^n f(\mathbf{x}_i)\right). \tag{24}$$

According to Assumption 2, we have the largest value of elements in $\mathbf{K}^{(p)}$ is $b$. Now, we consider $\tilde{\mathbf{K}}^{(p)} = \mathbf{K}^{(p)} + \alpha\mathbf{I} - \lambda\mathbf{GG}^T$. Since the diagonal elements of $\mathbf{GG}^T$ are 1, given any $\mathbf{x}_i$, we have that $\Phi_{\boldsymbol{\gamma}}^T(\mathbf{x}_i)\Phi_{\boldsymbol{\gamma}}(\mathbf{x}_i) = \sum_{p=1}^m \gamma_p^2 \Phi_p^T(\mathbf{x}_i)\Phi_p(\mathbf{x}_i) \le b + \alpha - \lambda$. Given two different instances $\mathbf{x}_i$ and $\mathbf{x}_j$, if they belong to the same protected group, we have that the $(i,j)$-th element in $\mathbf{GG}^T$ is 1, and thus $\Phi_{\boldsymbol{\gamma}}^T(\mathbf{x}_i)\Phi_{\boldsymbol{\gamma}}(\mathbf{x}_j) = \sum_{p=1}^m \gamma_p^2 \Phi_p^T(\mathbf{x}_i)\Phi_p(\mathbf{x}_j) \le b - \lambda$. If $\mathbf{x}_i$ and $\mathbf{x}_j$ belong to different protected groups, we have that the $(i,j)$-th element in $\mathbf{GG}^T$ is 0, and thus $\Phi_{\boldsymbol{\gamma}}^T(\mathbf{x}_i)\Phi_{\boldsymbol{\gamma}}(\mathbf{x}_j) = \sum_{p=1}^m \gamma_p^2 \Phi_p^T(\mathbf{x}_i)\Phi_p(\mathbf{x}_j) \le b$.

Then, notice that

$$\min_{\mathbf{y}\in\{\mathbf{e}_1,\dots,\mathbf{e}_c\}} \|\Phi_{\boldsymbol{\gamma}}(\mathbf{x}_i) - \mathbf{My}\|_{\mathcal{H}}^2 = \min\left\{\|\Phi_{\boldsymbol{\gamma}}(\mathbf{x}_i) - \mathbf{m}_1\|_{\mathcal{H}}^2, \cdots, \|\Phi_{\boldsymbol{\gamma}}(\mathbf{x}_i) - \mathbf{m}_c\|_{\mathcal{H}}^2\right\}, \tag{25}$$

where $\mathbf{m}_k$ denotes the $k$-th cluster centroid. Next, we denote $a_i = |\pi_k \cap \mathcal{G}_i|$ to represent the number of instances in all protected groups in cluster $\pi_k$. We have:

$$\left\| \Phi_{\boldsymbol{\gamma}}\left(\mathbf{x}_i\right) - \mathbf{m}_k \right\|_{\mathcal{H}}^2 = \left\| \Phi_{\boldsymbol{\gamma}}(\mathbf{x}_i) - \frac{1}{|\pi_k|} \sum_{j \in \pi_k} \Phi_{\boldsymbol{\gamma}}\left(\mathbf{x}_j\right) \right\|_{\mathcal{H}}^2$$

$$\leq 2 \left( \Phi_{\boldsymbol{\gamma}}^T(\mathbf{x}_i) \Phi_{\boldsymbol{\gamma}}(\mathbf{x}_i) + \frac{1}{|\pi_k|^2} \sum_{\mathbf{x}_p \in \pi_k} \sum_{\mathbf{x}_q \in \pi_k} \Phi_{\boldsymbol{\gamma}}(\mathbf{x}_p)^T \Phi_{\boldsymbol{\gamma}}(\mathbf{x}_q) \right)$$

$$\leq 2 \left( (b + \alpha - \lambda) + \frac{(|\pi_k|)(b + \alpha - \lambda) + \left( \sum_{i=1}^t a_i^2 - |\pi_k| \right)(b - \lambda) + \left( |\pi_k|^2 - \sum_{i=1}^t a_i^2 \right) b}{|\pi_k|^2} \right)$$

$$= 2 \left( (b + \alpha - \lambda) + \frac{|\pi_k|\alpha - \lambda \sum_{i=1}^t a_i^2 + b|\pi_k|^2}{|\pi_k|^2} \right)$$

$$\leq 2 \left( b + \alpha - \lambda + b - \frac{\lambda}{t} + \frac{\alpha}{|\pi_k|} \right)$$

$$\leq 4(b + \alpha) - 2 \left( 1 + \frac{1}{t} \right) \lambda \tag{26}$$

The second to last inequality holds due to the Cauchy-Schwarz inequality $\sum_{i=1}^t a_i^2 \geq \frac{\left( \sum_{i=1}^t a_i \right)^2}{t}$ and $\sum_{i=1}^t a_i = |\pi_k|$. Therefore, we have:

$$0 \leq f(x_i) \leq 4(b + \alpha) - 2 \left( 1 + \frac{1}{t} \right) \lambda. \tag{27}$$

According to the Theorem 3.1 in [30], by utilizing McDiarmid's inequality, we have that for any $\delta \geq 0$, with probability at least $1 - \delta$, for all $f \in \mathcal{F}$, the following inequality holds:

$$\mathbb{E}[f(\mathbf{x})] - \frac{1}{n} \sum_{i=1}^n f(\mathbf{x}_i) \leq 2\mathfrak{R}_n(\mathcal{F}) + \left( 4(b + \alpha) - 2 \left( 1 + \frac{1}{t} \right) \lambda \right) \sqrt{\frac{\log(1/\delta)}{2n}}, \tag{28}$$

where:

$$\mathfrak{R}_n(\mathcal{F}) = \frac{1}{n} \mathbb{E} \left[ \sup_{f \in \mathcal{F}} \sum_{i=1}^n \sigma_i f(\mathbf{x}_i) \right], \tag{29}$$

represents the Rademacher complexity of $\mathcal{F}$ [45]. $\sigma_1, \ldots, \sigma_n$ are Rademacher random variables uniformly distributed on $\{-1, 1\}$.

Next, we introduce the Gaussian complexity to provide an upper bound for $\mathfrak{R}_n(\mathcal{F})$ [5]:

$$\mathfrak{G}_n(\mathcal{F}) = \frac{1}{n} \mathbb{E} \left[ \sup_{f \in \mathcal{F}} \sum_{i=1}^n \beta_i f(\mathbf{x}_i) \right], \tag{30}$$

where $\beta_1, \ldots, \beta_n$ are Gaussian random variables with zero mean and unit standard deviation. To bound the Rademacher complexity, we need the following two lemmas:

**Lemma 2** *[23]* $\mathfrak{R}_n(\mathcal{F}) \leq \sqrt{\frac{\pi}{2}} \mathfrak{G}_n(\mathcal{F})$.

**Lemma 3** *[23] Let $G_f = \sum_{i=1}^n \beta_i G(\mathbf{x}_i, f)$ and $H_f = \sum_{i=1}^n \beta_i H(\mathbf{x}_i, f)$ be two zero-mean separable Gaussian processes. If for all $f_1, f_2 \in \mathcal{F}$,*

$$\mathbb{E} \left[ \left( G_{f_1} - G_{f_2} \right)^2 \right] \leq \mathbb{E} \left[ \left( H_{f_1} - H_{f_2} \right)^2 \right], \tag{31}$$

*then we have:*

$$\mathbb{E} \left[ \sup_{f \in \mathcal{F}} G_f \right] \leq \mathbb{E} \left[ \sup_{f \in \mathcal{F}} H_f \right]. \tag{32}$$

In our setting, we define:

$$G\left(\mathbf{x}_i, f\right) = G_{\mathbf{M},\boldsymbol{\gamma}} \triangleq \sum_{i=1}^n \beta_i \left(\min_{\mathbf{y}\in\{\mathbf{e}_1,\dots,\mathbf{e}_k\}} \|\Phi_{\boldsymbol{\gamma}}\left(\mathbf{x}_i\right) - \mathbf{M}\mathbf{y}\|_{\mathcal{H}}^2\right). \tag{33}$$

Next, we aim to find $H_f$ (i.e., $H_{\mathbf{M},\boldsymbol{\gamma}}$) such that:

$$\mathbb{E}_\beta\left[\left(G_{\mathbf{M}_1,\boldsymbol{\gamma}_1} - G_{\mathbf{M}_2,\boldsymbol{\gamma}_2}\right)^2\right] \le \mathbb{E}_\beta\left[\left(H_{\mathbf{M}_1,\boldsymbol{\gamma}_1} - H_{\mathbf{M}_2,\boldsymbol{\gamma}_2}\right)^2\right]. \tag{34}$$

Specifically, for any $f_1, f_2 \in \mathcal{F}$, we have:

$$\left(\min_{\mathbf{y}} \|\Phi_{\boldsymbol{\gamma}_1}\left(\mathbf{x}_i\right) - \mathbf{M}_1\mathbf{y}\|_{\mathcal{H}}^2 - \min_{\mathbf{y}} \|\Phi_{\boldsymbol{\gamma}_2}\left(\mathbf{x}_i\right) - \mathbf{M}_2\mathbf{y}\|_{\mathcal{H}}^2\right)^2$$

$$\le \left(\max_{\mathbf{y}} \left\{\|\Phi_{\boldsymbol{\gamma}_1}\left(\mathbf{x}_i\right) - \mathbf{M}_1\mathbf{y}\|_{\mathcal{H}}^2 - \|\Phi_{\boldsymbol{\gamma}_2}\left(\mathbf{x}_i\right) - \mathbf{M}_2\mathbf{y}\|_{\mathcal{H}}^2\right\}\right)^2$$

$$= \left(\left(\|\Phi_{\boldsymbol{\gamma}_1}\left(\mathbf{x}_i\right)\|_{\mathcal{H}}^2 - \|\Phi_{\boldsymbol{\gamma}_2}\left(\mathbf{x}_i\right)\|_{\mathcal{H}}^2\right) + \max_{\mathbf{y}} \left\{2\left(\Phi_{\boldsymbol{\gamma}_2}^T\left(\mathbf{x}_i\right)\mathbf{M}_2 - \Phi_{\boldsymbol{\gamma}_1}^T\left(\mathbf{x}_i\right)\mathbf{M}_1\right)\mathbf{y} + \mathbf{y}^T\left(\mathbf{M}_1^T\mathbf{M}_1 - \mathbf{M}_2^T\mathbf{M}_2\right)\mathbf{y}\right\}\right)^2$$

$$\le \left(\left(\|\Phi_{\boldsymbol{\gamma}_1}\left(\mathbf{x}_i\right)\|_{\mathcal{H}}^2 - \|\Phi_{\boldsymbol{\gamma}_2}\left(\mathbf{x}_i\right)\|_{\mathcal{H}}^2\right) + \max_{\mathbf{y}} 2\left(\Phi_{\boldsymbol{\gamma}_2}^T\left(\mathbf{x}_i\right)\mathbf{M}_2 - \Phi_{\boldsymbol{\gamma}_1}^T\left(\mathbf{x}_i\right)\mathbf{M}_1\right)\mathbf{y} + \max_{\mathbf{y}}\mathbf{y}^T\left(\mathbf{M}_1^T\mathbf{M}_1 - \mathbf{M}_2^T\mathbf{M}_2\right)\mathbf{y}\right)^2$$

$$= \Bigg(\left(\|\Phi_{\boldsymbol{\gamma}_1}\left(\mathbf{x}_i\right)\|_{\mathcal{H}}^2 - \|\Phi_{\boldsymbol{\gamma}_2}\left(\mathbf{x}_i\right)\|_{\mathcal{H}}^2\right) + \max_{\mathbf{y}} 2\sum_{r=1}^c y_r\left(\Phi_{\boldsymbol{\gamma}_2}^T\left(\mathbf{x}_i\right)\mathbf{M}_2 - \Phi_{\boldsymbol{\gamma}_1}^T\left(\mathbf{x}_i\right)\mathbf{M}_1\right)\mathbf{e}_r$$

$$+ \max_{\mathbf{y}} \sum_{r,s=1}^c y_r y_s \mathbf{e}_r^T\left(\mathbf{M}_1^T\mathbf{M}_1 - \mathbf{M}_2^T\mathbf{M}_2\right)\mathbf{e}_s\Bigg)^2$$

$$\le 4\left(\|\Phi_{\boldsymbol{\gamma}_1}\left(\mathbf{x}_i\right)\|_{\mathcal{H}}^2 - \|\Phi_{\boldsymbol{\gamma}_2}\left(\mathbf{x}_i\right)\|_{\mathcal{H}}^2\right)^2 + 2\left(\max_{\mathbf{y}} 2\sum_{r=1}^c y_r\left(\Phi_{\boldsymbol{\gamma}_2}^T\left(\mathbf{x}_i\right)\mathbf{M}_2 - \Phi_{\boldsymbol{\gamma}_1}^T\left(\mathbf{x}_i\right)\mathbf{M}_1\right)\mathbf{e}_r\right)^2$$

$$+ 4\left(\max_{\mathbf{y}} \sum_{r,s=1}^c y_r y_s \mathbf{e}_r^T\left(\mathbf{M}_1^T\mathbf{M}_1 - \mathbf{M}_2^T\mathbf{M}_2\right)\mathbf{e}_s\right)^2$$

$$\le 4\left(\|\Phi_{\boldsymbol{\gamma}_1}\left(\mathbf{x}_i\right)\|_{\mathcal{H}}^2 - \|\Phi_{\boldsymbol{\gamma}_2}\left(\mathbf{x}_i\right)\|_{\mathcal{H}}^2\right)^2 + 8\sum_{r=1}^c\left(\left(\Phi_{\boldsymbol{\gamma}_2}^T\left(\mathbf{x}_i\right)\mathbf{M}_2 - \Phi_{\boldsymbol{\gamma}_1}^T\left(\mathbf{x}_i\right)\mathbf{M}_1\right)\mathbf{e}_r\right)^2$$

$$+ 4\sum_{r,s=1}^c\left(\mathbf{e}_r^T\left(\mathbf{M}_1^T\mathbf{M}_1 - \mathbf{M}_2^T\mathbf{M}_2\right)\mathbf{e}_s\right)^2. \tag{35}$$

The final two inequalities hold due to $(a+b+c)^2 \le 4a^2 + 2b^2 + 4c^2$, $\sum_{r=1}^c y_r = 1$, and $\sum_{r,s=1}^c y_r y_s = 1$. Therefore, combining Eq.(33) and Eq.(35), we have:

$$\mathbb{E}_\beta\left[\left(G_{\mathbf{M}_1,\boldsymbol{\gamma}_1} - G_{\mathbf{M}_2,\boldsymbol{\gamma}_2}\right)^2\right]$$

$$= \mathbb{E}_\beta\left[\left(\sum_{i=1}^n \beta_i\left[\min_{\mathbf{y}} \|\Phi_{\boldsymbol{\gamma}_1}\left(\mathbf{x}_i\right) - \mathbf{M}_1\mathbf{y}\|_{\mathcal{H}}^2 - \min_{\mathbf{y}} \|\Phi_{\boldsymbol{\gamma}_2}\left(\mathbf{x}_i\right) - \mathbf{M}_2\mathbf{y}\|_{\mathcal{H}}^2\right]\right)^2\right]$$

$$= \sum_{i=1}^n\left(\min_{\mathbf{y}} \|\Phi_{\boldsymbol{\gamma}_1}\left(\mathbf{x}_i\right) - \mathbf{M}_1\mathbf{y}\|_{\mathcal{H}}^2 - \min_{\mathbf{y}} \|\Phi_{\boldsymbol{\gamma}_2}\left(\mathbf{x}_i\right) - \mathbf{M}_2\mathbf{y}\|_{\mathcal{H}}^2\right)^2$$

$$\le \sum_{i=1}^n\Bigg[4\left(\|\Phi_{\boldsymbol{\gamma}_1}\left(\mathbf{x}_i\right)\|_{\mathcal{H}}^2 - \|\Phi_{\boldsymbol{\gamma}_2}\left(\mathbf{x}_i\right)\|_{\mathcal{H}}^2\right)^2 + 8\sum_{r=1}^c\left(\left(\Phi_{\boldsymbol{\gamma}_2}^T\left(\mathbf{x}_i\right)\mathbf{M}_2 - \Phi_{\boldsymbol{\gamma}_1}^T\left(\mathbf{x}_i\right)\mathbf{M}_1\right)\mathbf{e}_r\right)^2$$

$$+ 4\sum_{r,s=1}^c\left(\mathbf{e}_r^T\left(\mathbf{M}_1^T\mathbf{M}_1 - \mathbf{M}_2^T\mathbf{M}_2\right)\mathbf{e}_s\right)^2\Bigg]$$

$$= \mathbb{E}_\beta\left[\left(H_{\mathbf{M}_1,\boldsymbol{\gamma}_1} - H_{\mathbf{M}_2,\boldsymbol{\gamma}_2}\right)^2\right]. \tag{36}$$

Then, we obtain $H_{\mathbf{M},\boldsymbol{\gamma}}$ as follows:

$$H_{\mathbf{M},\boldsymbol{\gamma}} = 2\sum_{i=1}^n \beta_i \left\|\Phi_{\boldsymbol{\gamma}}^T\left(\mathbf{x}_i\right)\right\|_{\mathcal{H}}^2 + 2\sqrt{2}\sum_{i=1}^n\sum_{r=1}^c \beta_{ir}\Phi_{\boldsymbol{\gamma}}^T\left(\mathbf{x}_i\right)\mathbf{M}\mathbf{e}_r + 2\sum_{i=1}^n\sum_{r,s=1}^c \beta_{irs}\mathbf{e}_r^T\mathbf{M}^T\mathbf{M}\mathbf{e}_s. \tag{37}$$

To bound the expectation of $H_{\mathbf{M},\boldsymbol{\gamma}}$, we introduce the following Lemma from [31]:

**Lemma 4** *[31] Suppose that*

*1)* $(\mathbf{e}_r : 1 \leq r \leq c)$ *is an orthonormal basis of* $\mathbb{R}^c$ *;*

*2)* $\mathcal{M}$ *is the class of linear operators* $\mathbf{M} : \mathbb{R}^c \to H$ *with* $\|\mathbf{Me}_r\|_{\mathcal{H}} \leq \omega$

*3)* $(\mathbf{x}_i : 1 \leq i \leq n)$ *is a sequence in* $H, \|\mathbf{x}_i\|_{\mathcal{H}} \leq \mu$ *;*

*4)* $(\beta_{ir} : 1 \leq i \leq n, 1 \leq r \leq c)$ *and* $(\beta_{irs} : 1 \leq i \leq n, 1 \leq r, s \leq r)$ *are orthogaussian (independent and* $N(0,1)$ *) sequences.*

*Then the following three inequalities hold:*

$$\mathbb{E}_\beta \sup_{\mathbf{M} \in \mathcal{M}} \sum_{i=1}^{n} \sum_{r=1}^{c} \beta_{ir} \langle x_i, \mathbf{Me}_r \rangle \leq \omega \mu c \sqrt{n}, \tag{38}$$

$$\mathbb{E}_\beta \sup_{\mathbf{M} \in \mathcal{M}} \sum_{i=1}^{n} \sum_{r=1}^{c} \beta_{ir} \|\mathbf{Me}_r\|_{\mathcal{H}}^2 \leq \omega^2 c \sqrt{n}, \tag{39}$$

$$\mathbb{E}_\beta \sup_{\mathbf{M} \in \mathcal{M}} \sum_{i=1}^{n} \sum_{r,s=1}^{c} \beta_{irs} \langle \mathbf{Me}_r, \mathbf{Me}_s \rangle \leq \omega^2 c^2 \sqrt{n}, \tag{40}$$

*where* $\langle \cdot, \cdot \rangle$ *denotes the inner production.*

In our method, given an instance $\mathbf{x}_i$, we have that $\|\Phi_{\boldsymbol{\gamma}}(\mathbf{x}_i)\|_{\mathcal{H}} = \Phi_{\boldsymbol{\gamma}}^T(\mathbf{x}_i) \Phi_{\boldsymbol{\gamma}}(\mathbf{x}_i) \leq b + \alpha - \lambda$. Moreover, we also have $\|\mathbf{Me}_r\|_{\mathcal{H}} \leq \sqrt{b + \alpha - \frac{\lambda}{t}}$ according to Eq.(26). As a result, according to Lemma 4, the expectation of $H_{\mathbf{M},\boldsymbol{\gamma}}$ can be be bounded as follows,

$$\mathbb{E}_\beta \left[ \sup_{f \in \mathcal{F}} H_{\mathbf{M},\boldsymbol{\gamma}} \right]$$

$$= \mathbb{E}_\beta \left[ \sup_{f \in \mathcal{F}} 2 \sum_{i=1}^{n} \beta_i \|\Phi_{\boldsymbol{\gamma}}^T(\mathbf{x}_i)\|_{\mathcal{H}}^2 + 2\sqrt{2} \sum_{i=1}^{n} \sum_{r=1}^{c} \beta_{ir} \Phi_{\boldsymbol{\gamma}}^T(\mathbf{x}_i) \mathbf{Me}_r + 2 \sum_{i=1}^{n} \sum_{r,s=1}^{c} \beta_{irs} \mathbf{e}_r^T \mathbf{M}^T \mathbf{Me}_s \right]$$

$$\leq 2\mathbb{E}_\beta \left[ \sup_{f \in \mathcal{F}} \sum_{i=1}^{n} \beta_i \|\Phi_{\boldsymbol{\gamma}}^T(\mathbf{x}_i)\|_{\mathcal{H}}^2 \right] + 2\sqrt{2}\mathbb{E}_\beta \left[ \sup_{f \in \mathcal{F}} \sum_{i=1}^{n} \sum_{r=1}^{c} \beta_{ir} \Phi_{\boldsymbol{\gamma}}^T(\mathbf{x}_i) \mathbf{Me}_r \right] + 2\mathbb{E}_\beta \left[ \sup_{f \in \mathcal{F}} \beta_{irs} \mathbf{e}_r^T \mathbf{M}^T \mathbf{Me}_s \right]$$

$$\leq 2(b + \alpha - \lambda)\sqrt{n} + 2c\sqrt{2(b + \alpha - \lambda)\left(b + \alpha - \frac{\lambda}{t}\right)n} + 2c^2\left(b + \alpha - \frac{\lambda}{t}\right)\sqrt{n}$$

$$= 2\sqrt{n} \left[ (1 + c^2)(b + \alpha) - (1 + \frac{c^2}{t})\lambda + c\sqrt{2(b + \alpha - \lambda)\left(b + \alpha - \frac{\lambda}{t}\right)} \right] \tag{41}$$

Last, we can bound $\Re_n(\mathcal{F})$ with Lemma 2, Lemma 3, Eq.(29), Eq.(30), and Eq.(41):

$$\Re_n(\mathcal{F}) \leq \frac{1}{n}\sqrt{\pi/2}\, \mathbb{E}_\beta \left[ \sup_{f \in \mathcal{F}} G_{\mathbf{M},\boldsymbol{\gamma}} \right] \leq \frac{1}{n}\sqrt{\pi/2}\, \mathbb{E}_\beta \left[ \sup_{f \in \mathcal{F}} H_{\mathbf{M},\boldsymbol{\gamma}} \right]$$

$$\leq \frac{\sqrt{2\pi}}{\sqrt{n}} \left[ (1 + c^2)(b + \alpha) - (1 + \frac{c^2}{t})\lambda + c\sqrt{2(b + \alpha - \lambda)\left(b + \alpha - \frac{\lambda}{t}\right)} \right]. \tag{42}$$

Substituting Eq.(42) into Eq.(28), we finally obtain for any $\delta \geq 0$, with probability at least $1 - \delta$, for all $f \in \mathcal{F}$, the following holds:

$$\mathbb{E}[f(\mathbf{x})] \leq \frac{1}{n} \sum_{i=1}^{n} f(\mathbf{x}_i) + \frac{2\sqrt{2\pi}}{\sqrt{n}} \left[ (1 + c^2)(b + \alpha) - (1 + \frac{c^2}{t})\lambda + c\sqrt{2(b + \alpha - \lambda)\left(b + \alpha - \frac{\lambda}{t}\right)} \right]$$
$$+ \left( 4(b + \alpha) - 2\left(1 + \frac{1}{t}\right)\lambda \right) \sqrt{\frac{\log(1/\delta)}{2n}}. \tag{43}$$

This concludes the proof.

## C  Proof of Lemma 1

Since $\sigma_A$ is the smallest eigenvalue of $\mathbf{A}$, and $\sigma_B$ is the largest eigenvalue of $\mathbf{B}$, we have:

$$\mathbf{A} - \sigma_A \mathbf{I} \succeq 0, \tag{44}$$
$$\sigma_B \mathbf{I} - \mathbf{B} \succeq 0. \tag{45}$$

Summing up Eq.(44) and Eq.(45), we have:

$$\mathbf{A} - \mathbf{B} - (\sigma_A - \sigma_B)\mathbf{I} \succeq 0. \tag{46}$$

Considering the smallest eigenvalue of $\mathbf{A} - \mathbf{B}$, denoting as $\sigma_{A-B}$, and its corresponding eigenvector $\mathbf{v}_{A-B}$, we have $(\mathbf{A} - \mathbf{B})\mathbf{v}_{A-B} = \sigma_{A-B}\mathbf{v}_{A-B}$. Multiplying the left-hand side of Eq.(46) with $\mathbf{v}_{A-B}^T$ and $\mathbf{v}_{A-B}$, we have

$$\mathbf{v}_{A-B}^T (\mathbf{A} - \mathbf{B} - (\sigma_A - \sigma_B)\mathbf{I}) \mathbf{v}_{A-B}$$
$$= \mathbf{v}_{A-B}^T (\sigma_{A-B} - (\sigma_A - \sigma_B)) \mathbf{v}_{A-B}$$
$$= (\sigma_{A-B} - (\sigma_A - \sigma_B)) \|\mathbf{v}_{A-B}\|_2^2. \tag{47}$$

Notice that $(\mathbf{A} - \mathbf{B} - (\sigma_A - \sigma_B)\mathbf{I})$ is positive semi-definite according to Eq.(46), which means $\mathbf{v}_{A-B}^T (\mathbf{A} - \mathbf{B} - (\sigma_A - \sigma_B)\mathbf{I}) \mathbf{v}_{A-B} \geq 0$. Therefore, $\sigma_{A-B} - (\sigma_A - \sigma_B) \geq 0$, and thus $\sigma_{A-B} \geq \sigma_A - \sigma_B \geq 0$. This means that $\mathbf{A} - \mathbf{B}$ is positive semi-definite, which concludes the proof.

## D  Statistical Information of Data Sets

We conduct experiments on benchmark data sets which are widely used in fair clustering, including D&S [2], HAR [3], Jaffe [29], MNIST-USPS [20], Credit Card [52] and K1b [53]. D&S is a human daily and sports activities data set including 8 participants. HAR is a human action recognition data set including 30 participants. In both D&S and HAR data sets, the data of each participant form a protected group. Jaffe is a face image data set. Following [20], the face images with the same expressions are put into a protected group. MNIST-USPS is an image data set containing images of handwritten digits from the subsets of MNIST and USPS data sets. Following [20], we randomly sample 2000 images from MNIST to form one protected group and randomly sample 1800 images from USPS to form the other protected group. Credit card is a data set that describes the customers' default payments and the data of males and females form two protected groups respectively. K1b is a text data set. Following [48], we randomly assign each text to a protected group with a Bernoulli distribution whose $p = 0.5$ to form two protected groups. Details of the data sets are shown in Table 3.

## E  Introduction of Compared Methods

To show the effectiveness of our method on clustering performance and fairness, we compare our method with some state-of-the-art fair clustering and multiple kernel k-means methods, including:

- **SpFC** [18], which integrates fairness constraints into the Laplacian matrix of a graph.
- **VFC** [58], which is a universal variational fair clustering framework.

Table 3: Description of the data sets.

| Data sets | # of Instances | # of Features | # of Cluster | Protected Groups |
|---|---|---|---|---|
| D&S | 9120 | 5625 | 19 | Person Identity (8) |
| HAR | 10299 | 561 | 6 | Person Identity (30) |
| MNIST-USPS | 3800 | 256 | 10 | Source of images (2) |
| Jaffe | 213 | 676 | 10 | Expression (7) |
| Credit Card | 5000 | 22 | 5 | Gender (2) |
| K1b | 2340 | 21839 | 6 | Synthetic Binary (2) |

- **FFC** [34], which is a three-stage fair clustering method based on k-means method.
- **ONKC** [25], which is an optimal neighborhood kernel clustering algorithm to enhance the representability of the optimal kernel.
- **MKCSS** [55], which is a simple yet effective neighbor-kernel-based MKC algorithm to consider the intrinsic neighborhood structure among base kernels.
- **DPMKKM** [42], which is a novel discrete multiple kernel k-means by directly solving the clustering indicator matrix.
- **LFLKA** [51], whihc is a simple late fusion multiple kernel clustering with local kernel alignment maximisation approach.
- **EMKC** [38], which is effective multiple kernnel k-means by introducing spectral perturbation theory to laplacian matrix.
- **OSLR** [47], which is a one stage multiple kernel k-means by refining shifted laplacian matrix.
- **ASLR** [46], which is a effective multiple kernel k-means by reconstructing the laplacian matrix.
- **CSAMKC** [56], which is a fast multiple kernel k-means by adopting a novel sampling strategy to improve the performance of MKC.
- **FAMKKM** [41], which is fast and innovative multiple kernel k-means by incorporating two approximated partition matrices instead of the original individual partition matric for each base kernel.

# F   Efficiency Results

Figures 3 and 4 show the convergence curves of FKKM and FMKKM, respectively. We can see that our methods converge very fast and they often converge within 5 iterations.

Figures 5 and 6 show the running time of all methods on single kernel setting and multiple kernel setting, respectively. For better comparison, we report the logarithm of the time (in seconds). From Figures 5 and 6, we can see that our FKKM and FMKKM are faster than or at least comparable with many state-of-the-art methods, which well demonstrates the efficiency of our methods.

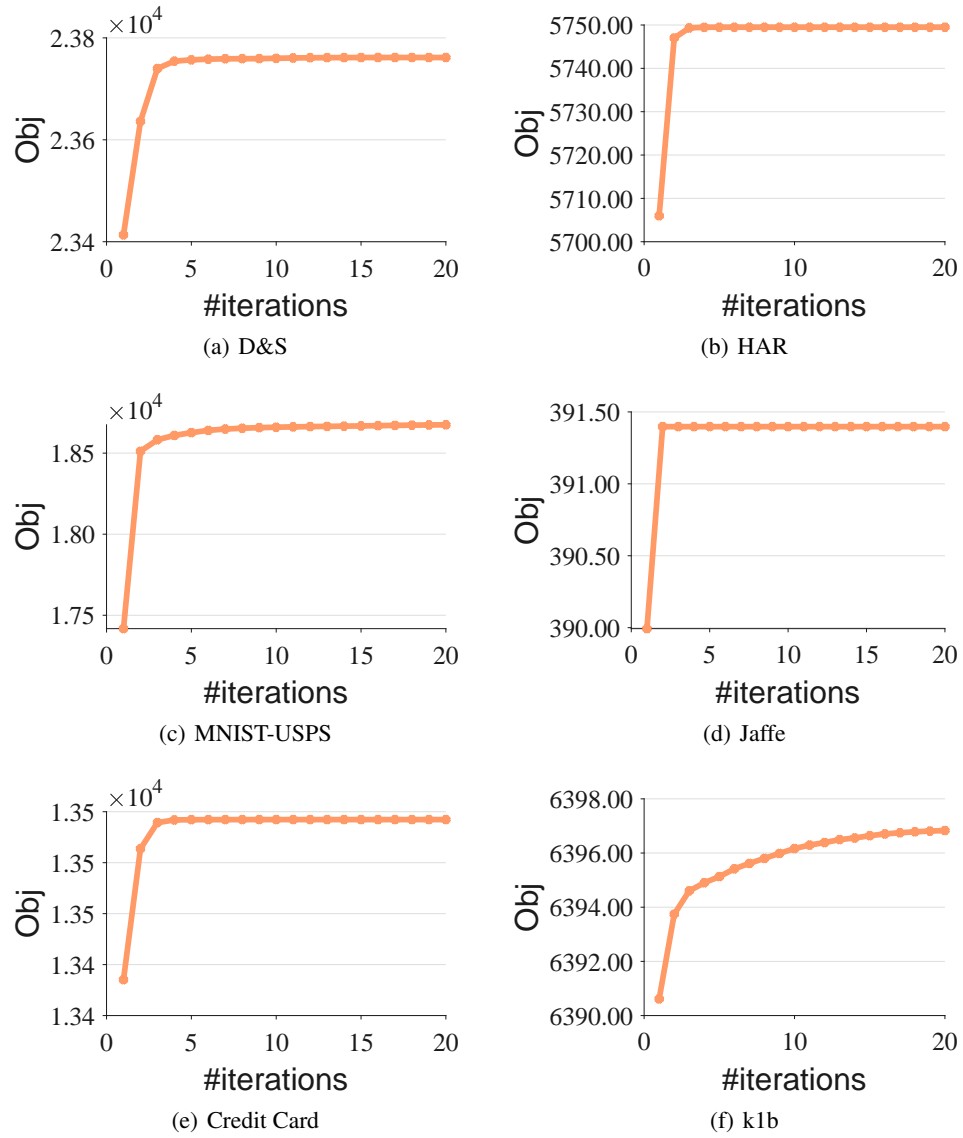

Figure 3: Convergence curves of all data sets on single kernel setting.

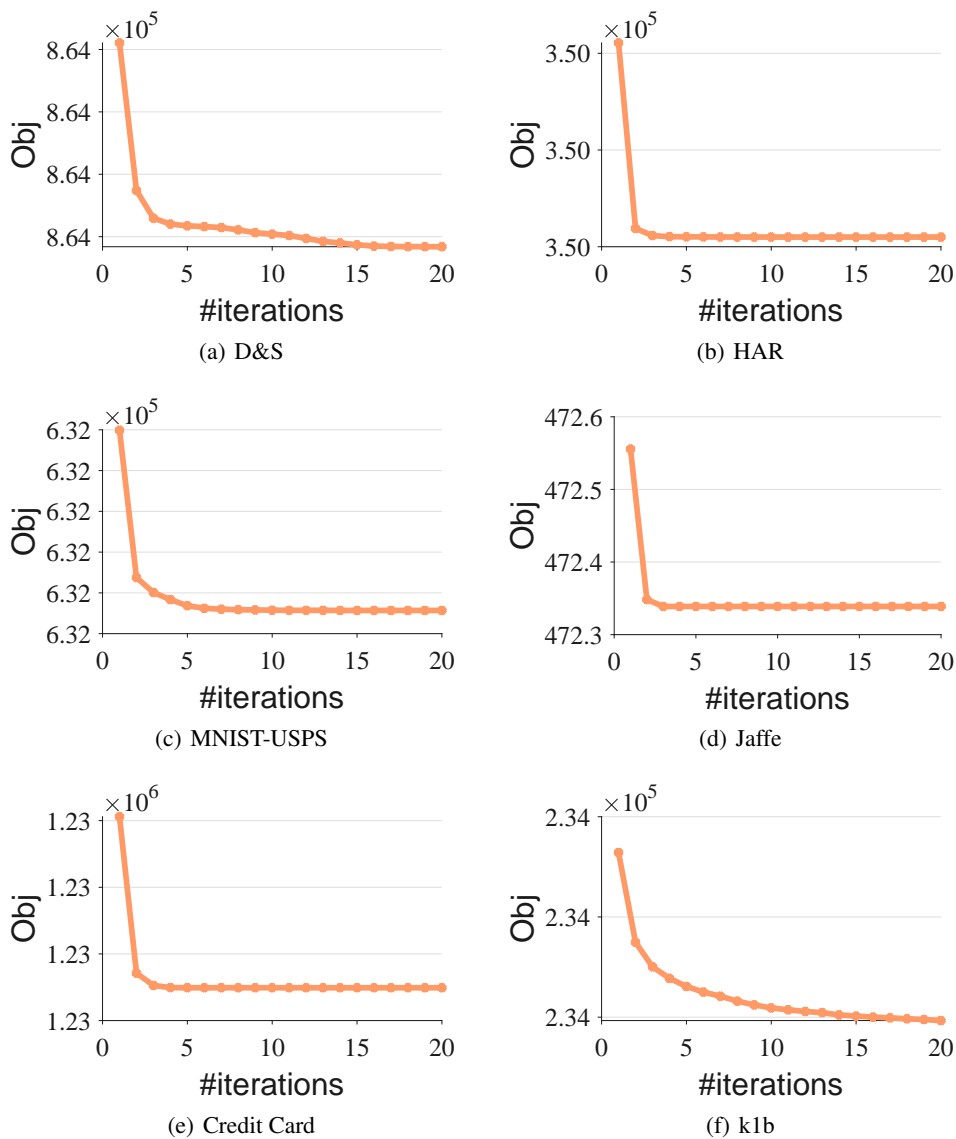

Figure 4: Convergence curves of all data sets on multiple kernel setting.

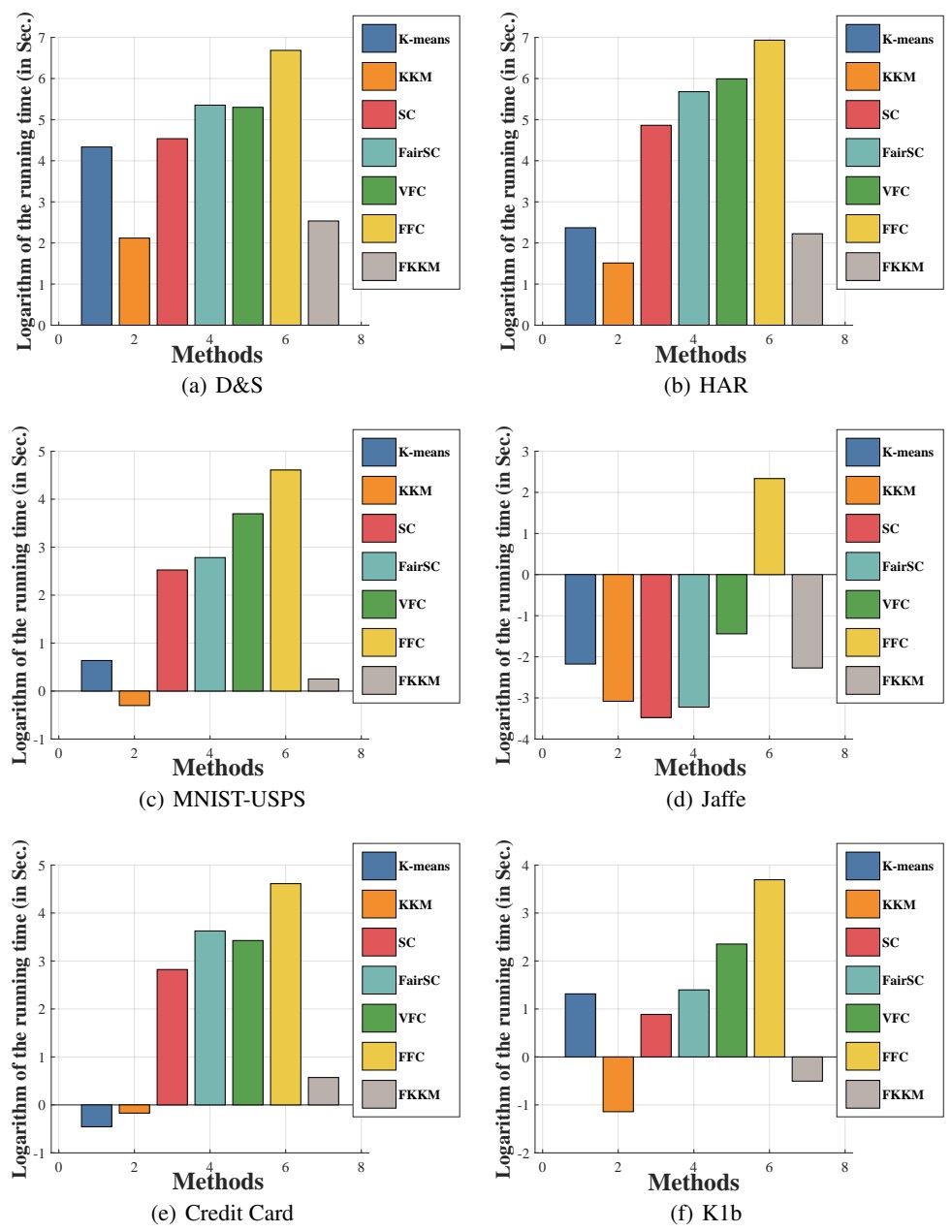

Figure 5: Running time of all methods on the single kernel setting.

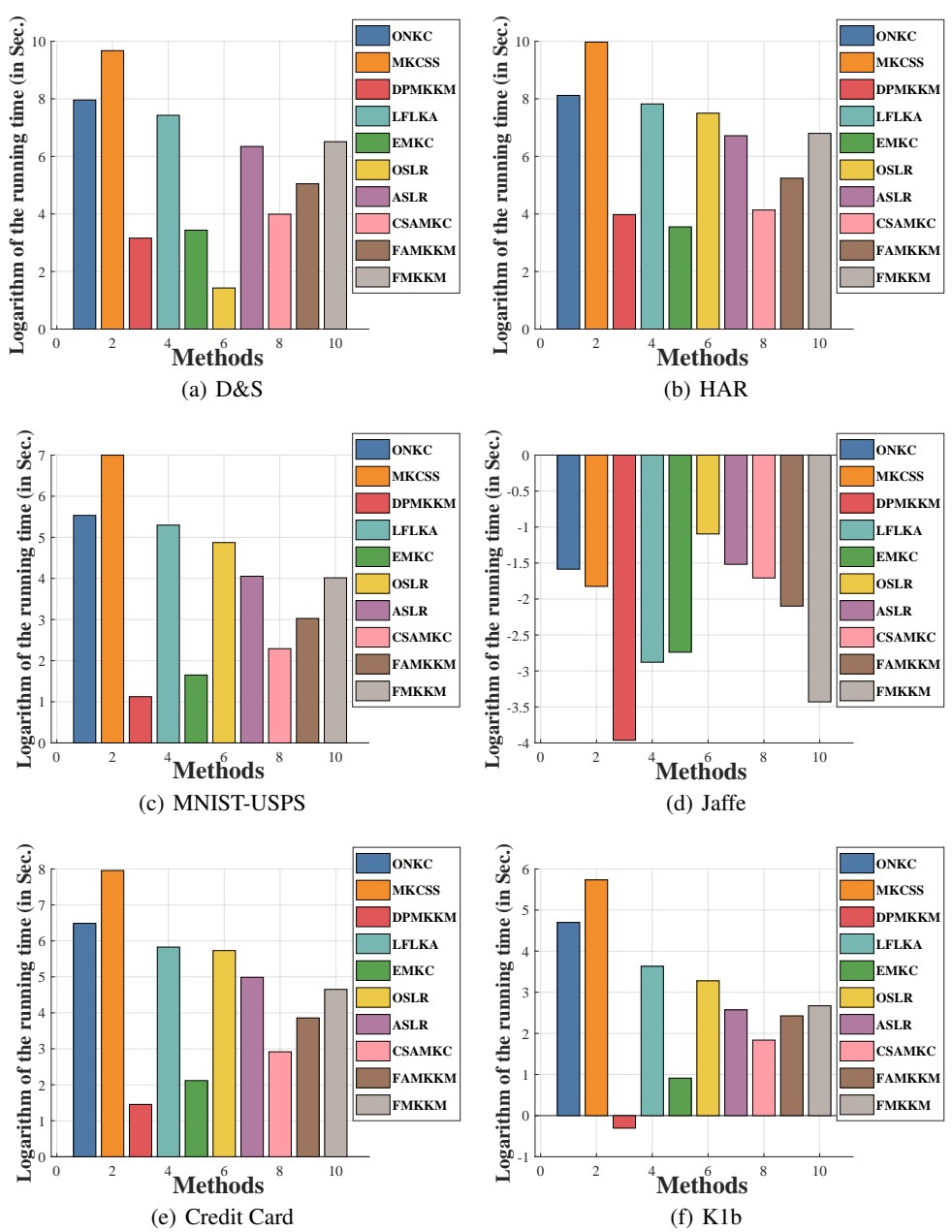

Figure 6: Running time of all methods on the multiple kernel setting.

