# OpenReview forum: "Fair Kernel K-Means: from Single Kernel to Multiple Kernel"
_NeurIPS.cc/2024/Conference — NeurIPS 2024 poster_

### Official Review · Reviewer_px1V · 2024-07-08

**Soundness:** 4
**Presentation:** 3
**Contribution:** 3
**Rating:** 7
**Confidence:** 5

**Summary:**

This paper focuses on the fairness in the kernel k-means. It designs a new fairness regularized term, which has the same form as the kernel k-means. Then it plugs this term into the kernel k-means and extends it to the multiple kernel k-means. Some theoretical analyses are provided to help to tune the hyper-parameters. The experimental results show the effectiveness of the proposed methods.

**Strengths:**

1. A novel fairness regularized term is proposed. It has the same form as the kernel k-means and can be seamlessly plugged into the kernel k-means framework.
2. Some theoretical analyses are provided. The paper is technically sound.
3. The paper also provides a strategy to tune the trade-off hyper-parameter $\lambda$ based on the generalization error bound. This makes the proposed method easily applicable to new data.
4. The experimental results on both the single kernel and multiple kernel settings are good, especially when comparing w.r.t. the fairness. It well demonstrates the effectiveness of the fairness regularized term.

**Weaknesses:**

1. The related work of fair clustering can be introduced in more detail.
2. As we know, time-consuming is one of the biggest problems in kernel methods. The proposed methods add a regularized term on the conventional kernel k-means and directly solve the discrete optimization problem. Will they increase time overhead?
3.  Although the paper discusses the trade-off between accuracy and fairness, I think more about the usage scenarios should be discussed, since the proposed methods are not as universal as conventional kernel k-means after all. For example, how about the scenarios that the accuracy contradicts to the fairness seriously? When should we use the proposed methods, and when should we use the conventional kernel k-means?

**Questions:**

Please see the Weaknesses.

**Limitations:**

The paper has analyzed the limitations.

---

> ### Author Rebuttal · Authors · 2024-08-06
>
> W1. We will revise the related work to introduce the fair clustering methods in more detail.
>
> W2. Since our method has the same form as the standard kernel k-means, we do not increase much overhead. In contrast, in our method, instead of using the eigenvalue decomposition which is used in conventional KKM and MKKM methods, we directly learn the discrete clustering result $Y$. This optimization only involves matrix multiplication, which is faster than conventional eigenvalue decomposition. We also conduct comparison experiments w.r.t. the running time. The results are shown in Figures 5 and 6 in the Appendix. The results show that our methods are faster than or at least comparable with other kernel methods.
>
> W3. We will discuss it in more detail in the revised version. The methods should be used in some applications involving humans which need fairness. For example, in the clustering of the customers of the banks, we wish to partition the customers into several groups to make the decisions for each individual. However, when doing the partition or making the decision, we should not consider gender, or it will cause sexism. In these scenarios considering the fairness when doing the partition, we can use the fair kernel k-means. Otherwise, in the cases that do not need fairness or without humans, we can use the standard clustering methods instead of the fair methods. If the accuracy contradicts to the fairness seriously, we should consider the fairness first or it may cause some bad social impact, such as sexism or other discriminations. Therefore, we should first guarantee fairness and select a relatively good clustering result in the fair results. Our strategy to choose $\lambda$ to make a trade-off between accuracy and fairness in Section 4 is also based on this rule.

---

> > ### Comment · Reviewer_px1V · 2024-08-12
> >
> > Thanks for the author's response, which has effectively addressed my concerns. I will keep my score on the paper.

---

### Official Review · Reviewer_t6xq · 2024-07-08

**Soundness:** 3
**Presentation:** 3
**Contribution:** 4
**Rating:** 7
**Confidence:** 5

**Summary:**

The authors design a novel fair kernel k-means method and a fair multiple kernel k-means method. The main part is the fairness regularization term. By minimizing this term, the optimal fairness, which is defined in Definition 1, can be achieved. The authors also derive the generalization error bound and discuss how to select $\lambda$ to decrease the bound. At last, the authors conduct the experiments by comparing with some state-of-the-art single kernel methods, fair clustering methods, and multiple kernel k-means methods. The results show that the proposed methods can achieve better fairness.

**Strengths:**

1.The fairness is an important issue in machine learning. Although kernel k-means have been widely studied in recent decades, fairness is seldom considered in these kernel methods. The paper considers the fairness for kernel methods, which can make contributions to the community.
2.The proposed methods are simple, elegant, and effective. After integrating the proposed fairness regularization term into kernel k-means, it is still a kernel k-means formula. The method only modifies the input kernel, but can achieve the fairness. This idea is interesting and novel.
3.The paper is well-motivated and solid. To my knowledge, the derivations are reasonable.
4.The paper is well-organized and easy to follow.
5.The experiments are sufficient and convincing. The ablation study of comparing with FKKM-f and FMKKM-f reveals the superiority of the fairness regularization term.

**Weaknesses:**

1.In Tables 1 and 2, there are many “0”s in the results of Bal. Is that normal? If so, why?
2.Figure 1 is a little confusing. Figure 1(b) is the result of FMKKM and should be fair. However, Figure 1(b) still seems not balanced enough. For example, when the cluster axis is 1, the instances is much more than the instances whose cluster axis is  20. More explanation should be provided.
3.Some symbols or notations e.g. $delta$ and $b$, are reused in different places for different purposes (see Section 4 and Section 5.1), which should be corrected.

**Questions:**

The paper calls $K+\alpha I-\lambda GG^T$ a fair kernel and claims that the methods only replace the traditional kernel with this fair kernel. However, traditional kernel k-means often needs the eigenvalue decomposition to learn a continual embedding and then discretize it to the final clustering results. The proposed methods directly learn the discrete results. My question is that, what about directly take the fair kernel $K+\alpha I-\lambda GG^T$ into the standard kernel k-means, instead of directly solving the new discrete problem? Can this still achieve the fairness?

---

> ### Author Rebuttal · Authors · 2024-08-06
>
> W1. Bal is a very strict evaluation metric that considers the worst case. Notice that $\mathrm{Bal}\left(\mathcal{C}\right)=\min_{k} \left(\frac{N_{k}^{\min}}{N_{k}^{\max}} \right)\in[0,1]$. As long as in one cluster, there are no instances of one protected group, according to its definition, Bal will be zero. That’s why there are many 0s in other methods. Therefore, the results are normal.
>
> W2. Sorry to cause you confusion. Fairness considers the balance of protected groups in each cluster instead of the balance of clusters. That means we should check the distribution along the protected group axis instead of the cluster axis. For example, although the instances in the cluster with axis 1 are much more than the cluster with axis 20, the 8 protected groups in the cluster with axis 1 or 20 are both balanced, which means the results are fair.
>
> W3. Thanks. We will revise these notations.
>
> Q1. As you suggested, we tried to take $K+\alpha I-\lambda GG^T$ into standard KKM, denoted as KKM-fair. The results are shown as follows:
> |          | |    K1b       |      |      |    |   Jaffe          |      |      |
> |----------|--------|-------|------|------|------------|-------|------|------|
> |          | ACC    | NMI   | Bal  | MNCE |ACC        | NMI   | Bal  | MNCE |
> | KKM      | 0.669  | 0.537 | 0.775| 0.989| 0.948      | 0.922 | 0    | 0.900|
> | KKM-fair | 0.573  | 0.444 | 0.818| 0.993| 0.985      | 0.974 | 0.333| 0.978|
>
> |          |  |   Credit|      |      |         |    D&S   |      |      |
> |----------|-------------|-------|------|------|------------|-------|------|------|
> |          | ACC         | NMI   | Bal  | MNCE | ACC        | NMI   | Bal  | MNCE |
> | KKM      | 0.381       | 0.140 | 0.550| 0.961| 0.552      | 0.602 | 0    | 0.530|
> | KKM-fair | 0.403       | 0.145 | 0.570| 0.968| 0.643      | 0.735 | 0    | 0.640|
>
> |          |   |    M-U|      |      |        |  HAR      |      |      |
> |----------|-------------|-------|------|------|------------|-------|------|------|
> |          | ACC         | NMI   | Bal  | MNCE | ACC        | NMI   | Bal  | MNCE |
> | KKM      | 0.396       | 0.421 | 0    | 0.003| 0.620      | 0.609 | 0    | 0.930|
> | KKM-fair | 0.401       | 0.358 | 0.045| 0.257| 0.765      | 0.691 | 0.009| 0.971|
>
> It shows that this kernel can indeed improve the fairness of standard KKM. Notice that the fairness of KKM-fair may be lower than our proposed FKKM. It is because that according to Theorem 1, the regularized term achieves fairness when we find a discrete $Y$ to minimize $tr(Y^TGG^TY(Y^TY)^{-1})$. If we use the traditional two-step method, even though we obtain an optimal embedding $H$, we cannot guarantee there exists a discrete $Y$ to make $H=Y(Y^TY)^{-1/2}$ in the second step. That’s why we propose a one-step method instead of the two-step method.

---

### Official Review · Reviewer_GAQw · 2024-07-16

**Soundness:** 3
**Presentation:** 3
**Contribution:** 2
**Rating:** 5
**Confidence:** 4

**Summary:**

This paper proposes a novel Fair Kernel K-Means (FKKM) framework to address the fairness issue in kernel k-means clustering. The authors introduce a fairness regularization term that can be seamlessly integrated into the kernel k-means objective function. They extend this approach to multiple kernel k-means, resulting in Fair Multiple Kernel K-Means (FMKKM). The paper provides theoretical analysis of the generalization error bound and a strategy for setting hyperparameters. Extensive experiments on both single kernel and multiple kernel settings demonstrate the effectiveness of the proposed methods in achieving fair clustering results while maintaining competitive clustering performance.

**Strengths:**

1. The paper introduces a new fairness regularization term that can be easily integrated into kernel k-means frameworks.
2. The authors provide thorough theoretical analysis, including proofs of the fairness optimization and generalization error bounds.
3. The method is extended from single kernel to multiple kernel settings, showing its adaptability.
4. The paper offers a strategy for setting hyperparameters based on theoretical analysis, making the method more accessible for practical use.
5. The authors conduct extensive experiments on multiple datasets, comparing their methods against various state-of-the-art approaches in both single and multiple kernel settings.
6. The proposed methods demonstrate fast convergence and competitive running times compared to existing methods.

**Weaknesses:**

1. The main weakness is that this paper does not discuss the use case extensively. In what cases people would use this kind of fair kernel based clustering method?
2. The evaluation is mainly about fairness and it is more convincing if the authors could compare the clustering performance. Usually incorporating fairness would cause the degeneration on the original clustering performance. A significant improvement on fairness and a small degeneration on clustering can better validate the effectiveness of the proposed method.

**Questions:**

1. How does the proposed method perform on very large-scale datasets? Are there any scalability issues?
2. Have the authors considered extending the fairness concept to other clustering algorithms beyond kernel k-means?
3. How sensitive are the proposed methods to the choice of initial kernels in the multiple kernel setting?
4. Can the proposed fairness regularization term be adapted to other machine learning tasks beyond clustering?

**Limitations:**

1.  As mentioned in the conclusion, the method requires pre-defined protected groups, which may not always be available or appropriate in all scenarios.
2. While the method often achieves good performance on both metrics, there might be cases where improving fairness significantly impacts clustering quality.
3. The approach is specifically designed for kernel-based clustering, which may limit its applicability to non-kernel-based clustering algorithms.
4. The paper focuses on a specific definition of fairness, and it's unclear how the method would perform under alternative fairness criteria.

---

> ### Author Rebuttal · Authors · 2024-08-06
>
> W1. The methods can be used in some applications involving humans which need fairness. For example, in the clustering of the customers of the banks, we wish to partition the customers into several groups to make the decisions for each individual. However, when doing the partition or making the decision, we should not consider gender, or it will cause sexism. In these scenarios considering the fairness when doing the partition, we can use the fair kernel k-means. Or otherwise, in the cases that do not need fairness or without humans, we can use the standard clustering methods instead of the fair methods. We will add more detailed discussion in the revised version.
>
> W2. In Tables 1 and 2, we compare with other methods w.r.t. ACC, NMI, Bal, and MNCE. The Bal and MNCE are for fairness, and the ACC and NMI are for clustering performance. As you said, incorporating fairness may cause the degeneration of the original clustering performance, which is also discussed in Section 4 and can be observed in our experiments of Parameter Study (i.e., Figure 2). To address this problem, we design a parameter selection method to choose the appropriate $\lambda$ for the regularized term in Section 4. We gradually enlarge $\lambda$ from 0, set $\alpha = \lambda ∗ |G_{max}|$, and observe the fairness metric. If it gets stable good fairness, we stop enlarging $\lambda$ and set $\lambda$ as the current value. This strategy does not need the ground truth, which is appropriate for unsupervised learning, and can obtain an as small as possible $\lambda$ to achieve a good fairness result. The comparison results also demonstrate this. Besides, another difference between our FKKM-f, FKKM and the conventional KKM is that our method is a one-step method that directly learns the final discrete clustering result $Y$ and other KKM methods are two-step methods that need to learn an embedding first, and then discretize the embedding to obtain the discrete result. In the two-step methods, the kernel k-means and the discretization post-processing are separated and when doing the discretization it cannot guarantee the clustering accuracy or fairness. We think this may be the other reason why our methods can outperform other methods w.r.t. the clustering accuracy.
>
> Q1. Yes. There is a scalability issue. The issue exists in the conventional KKM and MKKM methods, not only in our methods. We also have made some attempts to tackle the issue. For example, instead of using the eigenvalue decomposition used in conventional KKM and MKKM, we directly learn the discrete clustering result $Y$, which only involves matrix multiplication. It’s faster than conventional eigenvalue decomposition. We also conduct comparison experiments w.r.t. the running time. The results are shown in Figures 5 and 6 in Appendix. The results show that our methods are faster than or at least comparable with other kernel methods. Of course, some techniques for large scale kernel methods, such as [1], can be used to further improve the scalability and efficiency.
> [1] On the Consistency and Large-Scale Extension of Multiple Kernel Clustering. In IEEE TPAMI 2023.
>
> Q2. From the proof of Theorem 1, the proposed regularized term can achieve the fairness defined in Def. 1. Therefore, this term can be used in any clustering loss function involving the clustering indicator matrix $Y$. We use it in KKM and MKKM tasks because this term has the same form as KKM, and thus can be seamlessly integrated into these frameworks. We can also integrate it into other loss functions, such as kmeans and spectral clustering, but the formula may not be as elegant as that in KKM and MKKM.
>
> Q3. One main motivation of multiple kernel methods is that with different kernels which may have very different performance, the multiple kernel methods can provide a stable and robust result. Therefore, most MKC methods are insensitive to the choice of initial kernels. Since our method is a variation of standard MKKM, intuitively, it is also insensitive to the choice of the initial kernels.
>
> Q4. As explained in the response to your Q2, the proof of Theorem 1 doesn’t involve kernels or clustering. It means that Theorem 1 holds for any machine learning tasks. Therefore, the regularized term can be used in the tasks involving the learnable class indicator matrix $Y$, such as the classification, clustering, and some embedded feature selection methods using classification or clustering results.
>
> L2. Yes. As discussed in Section 4, too large weight for the regularized term may deteriorate the clustering accuracy. It can also be observed from our experiments of Parameter Study (i.e., Figure 2). To address this problem, we design a parameter selection method to choose the appropriate $\lambda$ for the regularized term in Section 4. The comparison results in Tables 1 and 2 with four metrics including the clustering performance and fairness also demonstrate this. Detailed explanation can be found in the answer to W2.
>
> L3. From the proof of Theorem 1, the proposed regularized term can achieve the fairness defined in Definition 1. Therefore, this term can be used in any clustering loss function involving the clustering indicator matrix $Y$. We use it in the KKM and MKKM tasks because this term has the same form as KKM and MKKM, and thus can be seamlessly integrated into these frameworks. We can also integrate it into other loss functions, such as kmeans and spectral clustering, but the formula may not be as elegant as that in KKM and MKKM.
>
> L4. Yes. Our regularized term is designed based on Definition 1, which is a widely used definition of fairness. Whether this term is effect under other definitions of fairness needs a more careful theoretical analysis.

---

> > ### Author Response · Authors · 2024-08-13
> >
> > Dear reviewer GAQw, Thank you for reviewing our paper. We hope our previous responses and revisions will meet your requirements. We are looking forward to your reply on the discussion stage. Thank you very much.

---

### Official Review · Reviewer_dRCy · 2024-07-19

**Soundness:** 3
**Presentation:** 2
**Contribution:** 2
**Rating:** 5
**Confidence:** 5

**Summary:**

The paper introduces a new framework called Fair Kernel K-Means (FKKM) aimed at addressing fairness issues in kernel K-means clustering. By incorporating a fairness regularization term, the method ensures fair data partitioning and avoids discrimination against specific groups. Additionally, the paper extends this method to a multi-kernel setting, proposing the Fair Multiple Kernel K-Means (FMKKM) method. Theoretical analyses and experimental validations demonstrate the effectiveness and superiority of the proposed methods in both single and multiple kernel clustering tasks.

**Strengths:**

1.The introduction of a fairness regularization term is a significant innovation that addresses the often overlooked issue of fairness in kernel K-means clustering.
2.Expanding the method to a multi-kernel setting (FMKKM) adds versatility and applicability to a broader range of clustering tasks.
3.The paper provides theoretical analysis, including a generalization error bound, and offers a strategy for hyperparameter settings, adding rigor to the proposed methods.
4.The methods are validated through experiments, showing their effectiveness in achieving fair clustering results.

**Weaknesses:**

1.While the paper introduces FKKM and FMKKM, it lacks a detailed discussion on their practical application and limitations. Real-world effectiveness and constraints are not thoroughly explored.
2.The experimental setup, including data selection and comparison with other methods, is not comprehensively described. There is a lack of in-depth analysis of the experimental results.
3.The paper does not detail the computational resources required for the experiments, such as CPU/GPU types, memory, and execution time, which are essential for ensuring reproducibility.
4.Although the paper proposes methods to address fairness, it does not thoroughly discuss how these methods perform in practical applications, especially in scenarios with noisy or inconsistent data.

**Questions:**

1.Are there specific technical challenges or deficiencies in the proposed fairness regularization term when applied to real-world, noisy, or unbalanced datasets? How does the method address these issues, and what are the potential impacts on its performance?
2.The paper combines fairness regularization with kernel K-means. How do these combinations offer substantial innovations beyond the sum of their parts? Are there specific examples where this combination significantly outperforms individual techniques?
3.Beyond the specific datasets tested, how does the proposed method generalize to other applications or domains? Can the authors provide examples or theoretical justifications for its broader applicability?
4.How does the new fairness regularization term quantify and balance the trade-off between clustering performance and fairness? Can the authors provide more detailed insights or metrics used to achieve this balance?
5.Given the computational complexity associated with fairness in clustering, how scalable is the proposed method for large-scale datasets? Can the authors provide a thorough analysis of the computational complexity and runtime performance compared to other state-of-the-art methods?
6.The paper mentions using multiple kernels. How does the proposed method ensure that the selected kernels contribute to fairness without compromising clustering performance? Are there specific strategies or metrics used to evaluate and enhance kernel diversity?
7.The experiments were conducted on specific benchmark datasets. How does the proposed method perform on diverse real-world datasets not included in the benchmark? Are there plans to validate the algorithm on more varied and complex datasets?
8.How sensitive is the proposed method to the choice of hyperparameters? Can the authors provide a sensitivity analysis to demonstrate the robustness of the method under different hyperparameter settings?
9.While the theoretical foundations are sound, how does the proposed method scale with extremely large datasets? Are there any practical implementations or optimizations that address the potential computational bottlenecks in real-world scenarios?
10.Given the complexity of fairness in clustering, how feasible is the method for real-time or near-real-time applications? Are there any real-world use cases where the proposed method has been successfully implemented and tested?

**Limitations:**

In the "Conclusion" section, the authors discuss the limitations of the work, including strong assumptions, the robustness of the results to the assumptions, the scope of the experiments, and factors that may affect the performance of the method. In addition, the authors discuss computational efficiency and scale issues.

---

> ### Author Rebuttal · Authors · 2024-08-06
>
> W1. Our methods can be used in applications involving humans which need fairness. For example, in bank system, we make decisions without considering the gender of customers to avoid sexism. In experiments, we use Credit Card data in this scenario. The data is to predict whether a customer will face default. It collects customers' information and partition the customers into 5 clusters: timely repayment, delayed repayment for 1, 2, 3, and 4 months, respectively. In this task, gender is the protected attribute to avoid sexism. We also use some other real-world data such as D&S and HAR. We’ll add more discussion in the revised version.
>
> W2. We’ll revise it to introduce the experimental setup and results further. We use the widely-used data in real-world fair clustering tasks, including D&S, HAR, Credit Card. Following previous work, we also use some synthetic data such as JAFFE, K1b, MNIST-USPS. The details are introduced in the paper. In single kernel setting, we compare some classical clustering methods, such as kmeans, KKM; and SOTA fair clustering methods, such as FairSC, VFC. In the multi-kernel setting, we compare some SOTA multi-kernel methods, such as ONKC, ASLR. We’ll further introduce them in the revised version.
>
> The experimental results show that our method outperforms others in fairness. It is often better or comparable on ACC and NMI. It obtains good trade-off between accuracy and fairness due to our  strategy of choosing an appropriate $\lambda$ for the regularized term in Section 4. We gradually enlarge $\lambda$ from 0 and set $\alpha = \lambda |G_{max}|$. When getting stable good fairness, we stop enlarging it. The strategy can obtain as small as possible $\lambda$ for good fairness. Besides, ours is a one-step method that directly learns the final discrete result and other KKM methods are two-step methods that learn an embedding first and then discretize it. In two-step methods, the two steps are separated. When discretization, it cannot guarantee accuracy or fairness. This is the reason why it outperforms other methods in accuracy.
>
> W3 & Q5. All experiments are conducted on a PC with an i7-12700 CPU and 32G RAM and repeated 10 times to report the average result. We’ve analyzed the time complexity in the last paragraph in Section 3.4.2, which is $O(n^2c)$. Other kernel methods need the time-consuming eigenvalue decomposition. Our method directly learns the discrete clustering result without eigenvalue decomposition. It only uses matrix multiplication, which is faster in practice. We’ve reported the execution time in Fig. 5 and 6 in Appendix. It shows that ours are often faster than or comparable with other SOTA methods.
>
> W4 & Q7. We use some real datasets such as Credit Card to show the application of the proposed method, showing its superiority. We will try more real-world data including noisy and inconsistent data in the future. Section 4 shows that in the worst case the generalization error is upper bounded, providing a guarantee of the method used on new datasets theoretically.
>
> Q1.Our main contribution is to propose a new fairness regularized term, and thus we focus on fairness instead of robustness or balance. Since our method has the same form as standard KKM, any robust or balanced techniques that can be used in KKM, can also be easily used in our method. Our term can also be plugged into other methods that can handle noisy or unbalanced data, to further improve fairness.
>
> Q2. We can easily plug the regularized term into KKM because it has the same form as KKM, showing its elegant structure. It’s one of our contributions. Our ablation study (in Tab. 1 and 2) by comparing with the versions without the term, denoted as FKKM-f and FMKKM-f, show that without this term the fairness is poor, demonstrating this combination outperforms individual techniques.
>
> Q3. In Section 4, we provide theoretical analysis of the generalization error bound. On any untested data, in the worst case, the generalization error of the proposed FMKKM is upper bounded. It provides a theoretical guarantee on unseen data. According to Theorem 1, the proposed term can be used in any applications involving class indicator $Y$.
>
> Q4 & Q8. According to Theorem 1, larger $\lambda$ causes fairer results. According to the generalization error bound in Eq.(21), large $\lambda$ increases the error bound, i.e., it may decrease the accuracy on unseen data. This is how the regularized term compromises accuracy and fairness theoretically. In practice, we show the sensitive curves of hyperparameter $\lambda$ in Fig. 2. Large $\lambda$ causes better fairness but worse accuracy, being consistent with theoretical analysis. We provide a strategy to select $\lambda$ without accessing the ground truth labels. We also mark the selected $\lambda$ in the curves, showing that our strategy often achieves good trade-off between accuracy and fairness.
>
> Q6. No matter in single or multiple kernel methods, there is always trade-off between the clustering performance and fairness, controlled by $\lambda$. In Section 4 we theoretically discuss the trade-off in multi-kernel setting. Based on the discussion, we provide a strategy to select $\lambda$. Since the main contribution is about fairness, we don’t control the diversity. Intuitively, considering diversity may improve performance. Since ours has the same form as KKM, any diversity term for KKM can also be used in our methods.
>
> Q9 & Q10. Kernel methods often have scalable issue. So our current version may also be hard to handle extremely large data or real-time applications. However, our implementation only uses matrix multiplication instead of eigenvalue decomposition. It can be easily parallelized for scalability. Moreover, since our formula has the same form as KKM, any scalable or speedup methods for KKM can also be used in ours to tackle the scalable issue, such as [1].
> [1] On the Consistency and Large-Scale Extension of Multiple Kernel Clustering. In TPAMI 2023.

---

> > ### Author Response · Authors · 2024-08-13
> >
> > Dear reviewer  dRCy,
> > Thank you for reviewing our paper. We hope our previous responses and revisions will meet your requirements. We are looking forward to your reply on the discussion stage.
> > Thank you very much.

---

> > ### Comment · Reviewer_dRCy · 2024-08-13
> > **Respond**
> >
> > Thank you for your answer which partly solve my problem.
> >
> > Regarding W4 & Q7, I want to know why you didn't try more real data, including noisy and inconsistent data, to verify the effectiveness of the algorithm in the experiments of this paper. Your reply told me that you originally intended to do so, but it was not shown in the paper. Why? I still have doubts about the case study.
> >
> > As you can see in AC's comment, I would be also interested to AC's question:
> > 1.about the correctness of Theorem 1. That mentioned equality comes from Cauchy-Schwarz in (11), which seems to hold under a broader condition (as long as the two vectors of added items are orthogonal).
> > 2. Why authors present the designed algorithm (18) in a seemingly unconventional way.

---

> > > ### Author Response · Authors · 2024-08-13
> > >
> > > Thanks for your responses. Our experiment follows previous works of fair clustering, and we use some public real-world data, such as Credit Card. Our reply just means that we tried some real-world data but not means we intend to use the noisy and inconsistent data. We admit that we do not consider the noisy and inconsistent data specifically. We appreciate your constructive suggestions and agree that using more noisy and inconsistent data would be better. We will try more data, especially noisy and inconsistent data.
> > >
> > > Sorry. We can not see AC's questions now. The following is the answers to the AC's two questions you mentioned.
> > > 1. The Cauchy-Schwarz Inequality is that for two vectors $\mathbf{a}$ and $\mathbf{b}$, we have $||\mathbf{a}||_2^2*||\mathbf{b}||_2^2\ge\<\mathbf{a},\mathbf{b}\>^2$, where $\<,.,\>$ denotes the inner production. The equation holds if and only if $\mathbf{a}=c\mathbf{b}$ for some $c$, which means $\frac{a_1}{b_1}=\frac{a_2}{b_2}=\cdots=\frac{a_n}{b_n}$. In our Theorem 1, $\mathbf{a}=[\frac{|\pi_1\cap\mathcal{G}_i|}{\sqrt{|\pi_1|}},\cdots,\frac{|\pi_c\cap\mathcal{G}_i|}{\sqrt{|\pi_c|}}]$ and $\mathbf{b}=[\sqrt{|\pi_1|},\cdots,\sqrt{|\pi_c|}]$. The equation in Eq.(11) holds, when $\frac{\frac{|\pi_1\cap\mathcal{G}_i|}{\sqrt{|\pi_1|}}}{\sqrt{|\pi_1|}}=\cdots=\frac{\frac{|\pi_c\cap\mathcal{G}_i|}{\sqrt{|\pi_c|}}}{\sqrt{|\pi_c|}}$, i.e., $\frac{|\pi_1\cap\mathcal{G}_i|}{|\pi_1|}=\cdots=\frac{|\pi_c\cap\mathcal{G}_i|}{|\pi_c|}$. Notice that, if $\frac{|\pi_1\cap\mathcal{G}_i|}{|\pi_1|}=\cdots=\frac{|\pi_c\cap\mathcal{G}_i|}{|\pi_c|}$, we have $\frac{|\pi_1\cap\mathcal{G}_i|}{|\pi_1|}=\cdots=\frac{|\pi_c\cap\mathcal{G}_i|}{|\pi_c|}=\frac{\sum_k{|\pi_k\cap\mathcal{G}_i|}}{\sum_k{|\pi_k|}}=\frac{|\mathcal{G}_i|}{n}$, which is shown in the paper. Therefore, the conditions shown in the proof of Theorem 1 are exactly the necessary and sufficient conditions of the equation holding.
> > >
> > > 2. Eq.(18) is the closed-form solution of $\gamma_p$ rather than our proposed algorithms. Due to the limited space, we do not show the algorithms in the main body of the paper, but we show the algorithms in the Appendix. The algorithms of FKKM and FMKKM are shown in Algorithms 1 and 2 in the Appendix. The differences between our algorithms and the standard KKM and MKKM are that 1) we need construction of fair kernel $\tilde{\mathbf{K}}=\mathbf{K}+\alpha\mathbf{I}-\lambda\mathbf{G}\mathbf{G}^T$, and 2) we optimize discrete $\mathbf{Y}$ row by row.

---

> > > > ### Comment · Reviewer_dRCy · 2024-08-13
> > > > **Response 2**
> > > >
> > > > Here is the questions:
> > > >
> > > > The concern about the correctness of Theorem 1. Its proof argues the equality in (12) holds when equlity in Row116, which implies minimizing the left-hand-side of (12) always results in  equlity in Row116. However, this equality comes from Cauchy-Schwarz in (11), which seems to hold under a broader condition (as long as the two vectors of added items are orthogonal). In other words,  equlity in Row116 may only be a sufficient but not necessary condition for the equality to hold, which means minimizing the left-hand-side of (12) does not always results in  equlity in Row116.
> > > >
> > > > Why authors present the designed algorithm (18) in a seemingly unconventional way. If the goal is to incorporate an existing fairness constraint, isn't it more common to first design a clear clustering objective (btw, what is the clustering objective of this algorithm?) with the fairness constraint, then kernelize it and finally optimize it or its proxy?

---

> > > > ### Comment · Reviewer_dRCy · 2024-08-13
> > > > **Response**
> > > >
> > > > With the clarification of case study and experiments, all my questions are mostly solved so I will improve my score.

---

> > > > > ### Author Response · Authors · 2024-08-14
> > > > >
> > > > > Thanks for your approval and the clarification of the AC's questions.
> > > > >
> > > > > For the Question 1, i.e., the correctness of Theorem 1. As said by AC, the equation in Row116 is a sufficient condition for the equality to hold. Therefore, we just show that it is also a necessary condition. To show it's a necessary condition, we need to show that if the equality in Eq.(11) holds, we can derive the equality in Row116. As discussed in above comments, the equality in Cauchy-Schwarz Inequality can result in  $\frac{|\pi_1\cap\mathcal{G}_i|}{|\pi_1|}=\cdots=\frac{|\pi_c\cap\mathcal{G}_i|}{|\pi_c|}$. So, we just need to show that this ratio can only be $\frac{|\mathcal{G}_i|}{n}$. Notice that if $\frac{a}{b}=\frac{c}{d}$, we have $\frac{a}{b}=\frac{c}{d}=\frac{a+c}{b+d}$. According to this, we have $\frac{|\pi_1\cap\mathcal{G}_i|}{|\pi_1|}=\cdots=\frac{|\pi_c\cap\mathcal{G}_i|}{|\pi_c|}=\frac{\sum_k{|\pi_k\cap\mathcal{G}_i|}}{\sum_k{|\pi_k|}}$. Since $\pi_k$ is a disjoint partition of all data, we have $(\pi_1\cap\mathcal{G}_i)\cup\cdots\cup(\pi_c\cap\mathcal{G}_i)=\mathcal{G}_i$ and $(\pi_p\cap\mathcal{G}_i)\cap(\pi_q\cap\mathcal{G}_i)=\emptyset$ for any $p,q$. Therefore, we have $\sum_k|\pi_k\cap\mathcal{G}_i|=|\mathcal{G}_i|$. Similarly, we have $\pi_1\cup\cdots\cup\pi_c=X$ and $\pi_p\cap \pi_q=\emptyset$, where $X$ is the set of all data, leading to $\sum_k|\pi_k|=n$. Therefore, we have  $\frac{|\pi_1\cap\mathcal{G}_i|}{|\pi_1|}=\cdots=\frac{|\pi_c\cap\mathcal{G}_i|}{|\pi_c|}=\frac{\sum_k{|\pi_k\cap\mathcal{G}_i|}}{\sum_k{|\pi_k|}}=\frac{|\mathcal{G}_i|}{n}$, which is the equation in Row116.
> > > > >
> > > > > For the second Question. Since our method is based on standard KKM, the clustering objective and kernelization have been done by standard KKM. In more detail, Eq.(1) in Section 2.1 is the clustering objective of standard KKM. It is kernelized in Eq.(2). Our contribution is to design a new fairness regularization term, which is shown in Eq.(8). Therefore, our final objective is to combine Eq.(2) and Eq.(8), which is shown in Eq.(14). Then we design Algorithm 1 to optimize Eq.(14), which first constructs the fair kernel $\tilde{\mathbf{K}}=\mathbf{K}+\alpha\mathbf{I}-\lambda\mathbf{G}\mathbf{G}^T$, and then optimize $\mathbf{Y}$ row by row. For the FMKKM, the original clustering objective is Eq.(4) in Section 2.1. We plug our regularized term (Eq.(8)) into it, leading to our final objective function Eq.(15). Then we design Algorithm 2 to optimize Eq.(15). We first construct fair kernels for each base kernel, and then iteratively optimizing $\mathbf{Y}$ and weight $\gamma_p$. I'm sorry to cause any confusion and will try to improve the presentation.

---

### Decision · Program_Chairs · 2024-09-25

**Decision:**

Accept (poster)

**Comment:**

This paper incorporates fairness into kernel K-means clustering, by designing a novel regularization term for clustering and proving it leads to fair clustering results based an existing fairness definition. A generalization error bound of the regularized clustering algorithm is derived and discussed, and experimental results on six data sets show the algorithm strikes a better fairness-accuracy balance than existing K-means clustering algorithms, their fair variants and kernel K-means clustering algorithms.

All reviewers commended the proposed regularization term, the theoretical analysis and convincing experimental results. I agree and think the regularization term is smartly designed and has a non-obvious connection to fair clustering (as stated in Theorem 1). The paper definitely carries a fair amount of merits.

On the other hand, reviewers suggest adding discussions on several things, including practical implications of fair kernel clustering (e.g. use cases), computational cost and related work. I agree and encourage authors to properly address them in revision or future studies.
Nevetheless, I judge these deficiencies do not outweigh the merits of the work.